# TPGPred: A Mixed-Feature-Driven Approach for Identifying Thermophilic Proteins Based on GradientBoosting

**DOI:** 10.3390/ijms252211866

**Published:** 2024-11-05

**Authors:** Cuihuan Zhao, Shuan Yan, Jiahang Li

**Affiliations:** 1Center for Synthetic and Systems Biology, School of Life Sciences, Tsinghua University, Beijing 100084, China; zhaocuihuan@mail.tsinghua.edu.cn; 2Institute of Public Safety Research, Department of Engineering Physics, Tsinghua University, Beijing 100084, China; 3School of Mathematical Sciences, Nankai University, Tianjin 300071, China

**Keywords:** thermophilic proteins, machine learning model, feature engineering, TPGPred

## Abstract

Thermophilic proteins maintain their stability and functionality under extreme high-temperature conditions, making them of significant importance in both fundamental biological research and biotechnological applications. In this study, we developed a machine learning-based thermophilic protein GradientBoosting prediction model, TPGPred, designed to predict thermophilic proteins by leveraging a large-scale dataset of both thermophilic and non-thermophilic protein sequences. By combining various machine learning algorithms with feature-engineering methods, we systematically evaluated the classification performance of the model, identifying the optimal feature combinations and classification models. Trained on a large public dataset of 5652 samples, TPGPred achieved an Accuracy score greater than 0.95 and an Area Under the Receiver Operating Characteristic Curve (AUROC) score greater than 0.98 on an independent test set of 627 samples. Our findings offer new insights into the identification and classification of thermophilic proteins and provide a solid foundation for their industrial application development.

## 1. Introduction

Thermophilic proteins are able to maintain their stability and biological activity under extreme heat conditions [1,2,3]. These proteins can function normally in environments with temperatures reaching 80 °C or even higher, exhibiting exceptional thermal stability and unique molecular adaptation mechanisms [4]. Research on thermophilic proteins is of significant scientific importance for understanding protein stability, folding mechanisms, and molecular adaptations at high temperatures. The in-depth exploration of these mechanisms not only deepens our understanding of life strategies in extreme environments but also provides new research directions in biochemistry, molecular biology, and structural biology [5,6,7]. Thermophilic proteins have broad prospects in industrial and biotechnology applications due to their excellent stability and activity [4,5]. For example, enzymes and proteins with high thermal stability are required in bioreactors operating under high temperatures and in the food processing, pharmaceuticals, and textile industries [8]. The use of thermophilic enzymes can enhance reaction rates, reduce contamination risks, and lower production costs, thereby improving the efficiency and sustainability of industrial processes. In the transition from laboratory research to industrial-scale production, the heat generated during microbial fermentation necessitates large cooling systems to remove excess heat and maintain optimal growth conditions for microorganisms. Identifying thermophilic proteins and using synthetic biology techniques to express suitable thermophilic proteins in industrial microbial chassis can transform engineered strains into thermotolerant variants, facilitating the cost-effective, large-scale production of target compounds. Therefore, discovering and designing new thermophilic proteins is of great practical significance for advancing biotechnology.

However, due to the diversity and complexity of protein sequences, systematically identifying and classifying thermophilic proteins remains a significant challenge [2,6]. Traditional methods primarily rely on wet-lab experiments, which are time-consuming and labor-intensive and are insufficient to handle the ever-growing volume of protein-sequence data in biological databases in a timely manner. With the development of high-throughput sequencing technologies, a vast number of unannotated protein sequences have accumulated in bioinformatics databases, urgently requiring efficient and accurate computational methods to screen for thermophilic proteins among them [9]. This is crucial for accelerating research and applications involving thermophilic proteins. *Thermus aquaticus*, an extreme thermophile capable of surviving at temperatures up to 80 °C, has become a cornerstone in biotechnology due to the discovery of its Taq DNA polymerase [3]. Its complete-genome sequencing has provided abundant information for studying thermophilic adaptations at the molecular level [1]. Nevertheless, the diversity and complexity of protein sequences still pose major challenges in systematically identifying and classifying thermophilic proteins. The rapid development of machine learning technologies offers new opportunities to address this problem [10]. By constructing machine learning-based predictive models and incorporating advanced feature-engineering methods, valuable patterns and features can be extracted from large amounts of protein-sequence data, enabling the efficient identification and classification of thermophilic proteins.

Numerous machine learning-based methods have been proposed and developed for the prediction and analysis of thermophilic proteins. Wang et al. [11,12] utilized amino acid composition and g-gap dipeptides as protein feature encodings, employing a support vector machine (SVM) as the classifier. Feng et al. [13] analyzed the physicochemical properties of amino acids using an SVM classifier, focusing on a dataset with 500 positive and 500 negative samples for the classification of thermophilic proteins. Meng et al. [14] employed seven types of protein features as input and used an SVM as the classifier. Tang et al. [15] used the frequency of amino acids and dipeptides as features and applied an SVM for the classification analysis of 1700 positive and negative samples. Guo et al. [16] analyzed ten protein-encoding features using an SVM-based model, evaluating their effectiveness in classification. Charoenkwan et al. [17,18] used 12 types of protein descriptors as input features and applied machine learning algorithms for feature analysis and classification. Zhao et al. [19] employed six types of protein-encoding descriptors as input, further processing them with a convolutional neural network (CNN) and bidirectional long short-term memory network (BiLSTM), and used a multi-layer perceptron (MLP) for classification. Ahmed et al. [20] utilized seven protein descriptors as features and employed MLP as the classifier for protein classification, relying solely on protein descriptors without integrating non-linear sequence feature-engineering methods. The aforementioned approaches primarily focus on the use of protein descriptors as the main feature set and analyze them using various machine learning methods. Pei et al. [21] directly used protein sequences as input, applying the transformer-embedding method within BERT for sequence feature processing, which is a pure sequence-based feature approach similar to Word2Vec and Doc2Vec. This model exclusively relies on a single-sequence feature-engineering method without performing a comparative analysis of different feature-engineering strategies. Li et al. [22] used machine learning models based on protein sequences to study the potential maximum tolerance temperature of thermophilic proteins, achieving an R² of 0.75. Ahmed et al. [23] performed statistical analysis from a structural biology perspective, examining secondary structure, hydrogen bonds, salt bridges, DHA (donor-hydrogen-acceptor) angles, and bond lengths. These studies primarily focus on the structural and applied aspects of thermophilic protein analysis.

Overall, current methodologies predominantly focus on single categories of features, such as protein-encoding or sequence-based characteristics. In the context of machine learning, features define the receptive field and scope through which the model perceives its target, serving as its primary source of information. Since machine learning methods are highly sensitive to the types of features employed, different feature-engineering strategies often align with distinct, optimal, machine learning algorithms. Rich and effective features are crucial for enhancing the performance of machine learning models. This study not only considers various feature-engineering methods based on protein sequences but also emphasizes model interpretability by incorporating protein descriptor-based feature-engineering techniques. The main contributions of this study to existing methodologies can be summarized as follows: (i) This study extensively employs protein-encoding features, introducing a total of 14 types of protein descriptors, which enhances the interpretability of the model’s results. (ii) The study incorporates a significant number of sequence-based features, using 11 different sequence feature-engineering methods for comparison and analysis, which further improves model performance. (iii) In terms of model selection, seven machine learning models are analyzed, with four models discussed in detail in the main text, and the results of the other three models are provided in the Appendix A. (iv) To further investigate the factors affecting model performance, this study explores four types of feature processing methods, three cross-validation techniques, and four feature-selection methods, conducting comparative analyses to assess their impact on model efficacy. Through comprehensive comparative analyses, this study achieves further improvements in model performance.

## 2. Results

### 2.1. Collection of a Comprehensive Thermophilic and Non-Thermophilic Protein Dataset

*Thermus aquaticus* is an extreme thermophilic bacterium capable of growing at temperatures as high as 80 °C, with its genome fully sequenced. To survive under such high-temperature conditions, the proteins within this bacterium must exhibit excellent thermostability. For this study, we downloaded the complete amino acid sequences of *Thermus aquaticus* proteins from NCBI, which were used as part of the training and test datasets, serving as our positive samples. Additionally, we employed the complete-genome sequences of *Halomonas* sp. TD01 sequenced in our laboratory as negative samples. The composition of the training and test datasets, including data sources, the distribution of thermophilic and non-thermophilic proteins, and the number of samples used, are summarized in Table 1. Specifically, the training set included 2213 proteins from extreme thermophiles and 3439 proteins from *Halomonas* sp. TD01 non-thermophiles. For the test set, we randomly selected 382 and 245 proteins from extreme thermophiles and non-thermophiles *Halomonas* sp. TD01, respectively, ensuring that these proteins were not included in the training set.

High-quality data are fundamental to effective model training and the evaluation of its generalization capabilities. They enable the more efficient testing and assessment of a model. During the dataset construction process, we excluded invalid amino acid sequences and conducted a thorough analysis of the selected sequences based on their composition, length, and symbols. Specifically, we filtered out sequences containing non-alphabetic characters, sequences with abnormal markers, invalid sequences, sequences shorter than 20 residues, and empty sequences. Additionally, we ensured that there was no overlap between the training and test datasets, and that the test set was randomly sampled from the complete dataset, preserving its independence from the training set. This rigorous data preparation ensured that the model’s performance was evaluated under conditions that reflect its ability to generalize to new, unseen data.

### 2.2. Construction of RNN Network Architecture

To conduct comparative analysis, in this study, we designed a recurrent neural network (RNN) architecture based on long short-term memory (LSTM) networks, featuring multiple stacked layers (Figure 1) [24,25]. The input layer receives the input data, where the feature dimension of the input sequence corresponds to the length of the feature column vector, with a total of two layers in the network. The first LSTM layer has a hidden dimension of 64, processing the input sequence and generating hidden states, which represent the network’s understanding of the input data at specific time points. These hidden states are updated at each time step, with sequential dependencies such that the hidden state of a previous time step influences the next. The second LSTM layer also has 64 hidden units, further processing the output from the first LSTM layer and ultimately generating the final hidden state, which is used to produce the final output. A fully connected layer takes the hidden state from the last time step of the second LSTM layer and passes it to a fully connected layer to generate the final output. The output dimension is 1, representing a scalar used for the final classification task. This network architecture is relatively shallow. Given the limited size of the input dataset, using a deeper network could result in severe overfitting, where the model performs well on the training data but exhibits poor generalization. Therefore, a shallower network structure was chosen for this study to balance performance and generalization, and to facilitate comparative analysis.

### 2.3. Analyzing Different Machine Learning Models with String Feature Engineering

The construction of model features determines the model’s ability to understand and interpret protein encoding. In this study, we combined various string-encoding methods commonly used in machine learning with 14 types of protein descriptors. String-encoding methods effectively capture the sequential relationships and micro-level statistical composition of amino acid sequences as strings. Protein descriptors, on the other hand, interpret amino acid sequences from the perspective of protein composition and encoding, offering a more intuitive and robust interpretability.

Due to differences in the network structures of each machine learning method, their abilities to recognize features on both micro and macro levels vary. Therefore, it was necessary to first identify the most suitable feature-engineering method for each machine learning model. We combined various machine learning algorithms, including Bagging, RandomForest, GradientBoosting, and ScikitRNN, with different feature-engineering techniques (e.g., grams, N-grams, hashing, LDA, LSA, PCA, T-SNE, Word2Vec, FastText, and Doc2Vec and BERT) to form multiple model combinations. To evaluate the performance of these combinations, we trained the models on the same feature dataset and validated them using the same test data. The performance of the models was comprehensively assessed using a series of common metrics, including Recall, Precision, Accuracy, F1-score, Matthews Correlation Coefficient (MCC), Area Under the Receiver Operating Characteristic Curve (AUROC), and Area Under the Precision–Recall Curve (AUPRC). The results demonstrated significant differences in performance across machine learning methods when different feature-engineering techniques were applied (Table 2). In the initial analysis phase, no specific model was designated as the baseline model for comparison. Instead, the focus was on identifying the most suitable feature-engineering methods for each machine learning model. The aim was to explore the optimal combinations of feature-engineering techniques and models without any preconceived preferences.

In the comparison of Bagging-based methods, the combination of Bagging with Doc2Vec performed exceptionally well, particularly in terms of Precision, F1-score, and MCC. This combination achieved an optimal balance between Recall and Precision, reducing both false positives and false negatives while delivering excellent classification performance. Furthermore, high AUROC and AUPRC scores indicated strong capability in distinguishing between positive and negative samples. Thus, the Bagging + Doc2Vec combination performed best across multiple metrics, making it an ideal choice for applications where a balance between Recall and Precision is required. For GradientBoosting-based methods, the GradientBoosting + HashingVectorizer combination also exhibited outstanding performance. The combination achieved a Precision and Specificity of 0.97 and 0.98, demonstrating strong ability in recognizing negative samples with minimal false positives. Its F1-score and MCC were also excellent, further validating its effectiveness in balancing Recall and Precision. The AUROC and AUPRC scores were 0.985 and 0.984, respectively, indicating robust performance in distinguishing positive from negative samples. Thus, the GradientBoosting + HashingVectorizer combination proved to be the best in this series, suitable for tasks requiring a balance between high Precision and high Recall, especially in scenarios with strict requirements for minimizing false positives. The RandomForest + FastText combination demonstrated a high Precision, indicating a very low false positive rate for positive samples. However, its Recall was only 0.80, limiting its ability to fully capture positive samples. Among the RandomForest-based methods, RandomForest + Doc2Vec was the best combination, showing strong performance across multiple metrics, including Precision, Recall, F1-score, MCC, and Specificity. It achieved a good balance between Recall and Precision, with particularly strong performance in Specificity. For ScikitRNN, the combination with CountVectorizer shows balanced performance in all indicators. ScikitRNN is used as a representative of recurrent machine learning models for comparison with other algorithms.

Based on the analysis, Doc2Vec consistently performed well across multiple machine learning methods, especially when combined with RandomForest and and Bagging, where it exhibited outstanding performance. Additionally, RandomForest and GradientBoosting demonstrated strong advantages in model stability and classification performance. Due to the differences in network structures and algorithm principles of each machine learning method, different feature-engineering methods were employed to optimize their training and testing results. According to the experimental results in Table 2, we identified the best-performing combinations: Bagging + Doc2Vec, RandomForest + Doc2Vec, ScikitRNN + CountVectorizer, and GradientBoosting + Word2Vec. These combinations will be further used in subsequent tests to compute and compare model performance.

### 2.4. Analyzing Different Machine Learning Models with Protein Descriptor Feature Engineering

Given the differences in feature processing and dimensionality analysis among different machine learning models, we further analyzed the individual impact of each protein descriptor on model performance. For multiple machine learning models, we sequentially removed protein descriptors and used the remaining descriptors to form different combinations for training and testing. We then analyzed the performance of each model in the absence of specific protein descriptors to identify the preference of each model for particular descriptors. The evaluation metrics included Recall, Precision, Accuracy, F1-score, MCC, Specificity, AUROC, and AUPRC. Due to the imbalance in the ratio of thermophilic to non-thermophilic proteins in the training data, we focused on key metrics such as Accuracy, AUROC, and AUPRC. AUROC reflects the overall performance of the model across different thresholds, while AUPRC is particularly suited for scenarios with imbalanced data, as it provides a better indication of the model’s performance at high Recall rates. Other metrics were used as supplementary references.

The number of features generated by different protein descriptors varied significantly. For example, GAAC (Global Amino Acid Composition) generates only five sub-features, while TPC (Tripeptide Composition) generates 8000 features. Directly comparing the feature counts of individual descriptors could introduce bias. Therefore, during the feature generation process, we employed a stepwise elimination method, where we removed each protein descriptor and its corresponding features, retaining only those generated by other descriptors. We then retrained and evaluated the machine learning models. This method effectively revealed the role of specific protein descriptors. If removing a descriptor improved the model’s performance on the test set (e.g., in terms of Accuracy, AUROC, AUPRC), it indicated that the descriptor had a negative impact on model training. Conversely, if removing a descriptor reduced the model’s predictive ability and the performance metrics dropped, it suggested that the descriptor played a positive role in model training, potentially providing important classification information.

Through the analysis of various protein descriptors under the Bagging_Doc2Vec model, the combination of GAAC_Bagging_Doc2Vec demonstrated outstanding performance across multiple metrics, achieving a Precision of 0.97, Accuracy of 0.84, Specificity of 0.98 (Figure 2a, Appendix A). These results indicate that GAAC_Bagging_Doc2Vec performs exceptionally well across several dimensions of classification, making it the most prominent performer among all tested combinations. Other combinations, such as DDE_Bagging_Doc2Vec and CTriad_Bagging_Doc2Vec, showed marginal improvements in certain metrics after removing the prefix, failing to reach the significant performance gains observed with GAAC. Comparing the performance differences before and after prefix removal in each combination further highlights the superiority of GAAC_Bagging_Doc2Vec, which saw an increase of more than 0.04 in both Specificity and Precision. This suggests that removing the prefix from protein descriptors significantly enhances classification performance. In contrast, combinations like TPC_Bagging_Doc2Vec experienced notable declines in performance, with Recall and Precision dropping by 0.66 and 0.27, respectively, indicating that this combination is heavily dependent on the prefix, and removing it severely compromises performance (Figure 2b, Appendix A). This result suggests that GAAC_Bagging_Doc2Vec is the best-performing combination within the Bagging_Doc2Vec model series, particularly excelling in tasks that require high Recall and Precision. Most combinations suffer from a decrease in Accuracy and F1-Score after prefix removal, while only some combinations show an improvement in performance. Therefore, the strategy for handling protein descriptor prefixes is crucial to the final classification performance and should be adjusted flexibly based on the task’s requirements.

The comparison of individual protein descriptors within the RandomForest_Doc2Vec model reveals both positive and negative impacts following their removal. Positive changes, such as with the AAC_RandomForest_Doc2Vec combination, demonstrate significant improvements in performance, particularly in Precision and F1-Score, after the removal of the protein descriptor prefix. This indicates that removing the descriptor prefix positively influences the model’s classification ability, especially in terms of balancing positive and negative samples. On the negative side, the TPC_RandomForest_Doc2Vec combination saw substantial drops in multiple metrics after prefix removal, indicating that this combination is highly dependent on the prefix, and its performance worsens significantly when the prefix is removed (Figure 3a, Appendix A).

By comparing the difference in performance metrics between models using full protein descriptors and those with prefixes removed, it was observed that after removing the prefix from AAC_RandomForest_Doc2Vec, MCC increased by 0.023683, Accuracy and F1-Score improved by 0.011, and Specificity rose by 0.028 (Figure 3b, Appendix A). Specificity remained at a high level, suggesting that this combination exhibits significant overall improvement after prefix removal, particularly in classification balance and Recall rate. This suggests that the model does not rely on the protein descriptor prefix and that its potential is further realized when the prefix is removed. It effectively distinguishes between positive and negative samples, and removing the prefix enhances its overall performance.

Overall, AAC_RandomForest_Doc2Vec is the best-performing combination in this series. The combination excels across multiple key metrics, with Precision reaching 0.9375 and Accuracy reaching 0.92185. Additionally, the Specificity of 0.96 highlights its exceptional ability to identify negative samples.

From the comparative analysis in Figure 4a (Appendix A), which examines the performance of GradientBoosting after the removal of individual protein descriptors, it is evident that the optimized model demonstrates a high sensitivity to changes in most performance metrics. The removal of these descriptors results in a relatively uniform decline in overall performance metrics, highlighting the critical role of each descriptor in maintaining the model’s predictive accuracy. Among the combinations, AAC_GradientBoosting_HashingVectorizer performed exceptionally well, particularly in terms of Precision (0.963134), Accuracy (0.929825), F1-Score (0.904762), and MCC (0.85345). These results suggest that this combination excels in Precision, overall Accuracy, and classification balance.

When comparing the performance differences after prefix removal to the baseline results using the full set of descriptors (Figure 4b, Appendix A), the combinations did not demonstrate significant improvement. However, among the various protein screening sequences, the GAAC_GradientBoosting_HashingVectorizer emerged as the best-performing combination. Most combinations exhibited negative impacts, with a marked decline in performance. This suggests that these combinations are more reliant on the presence of protein descriptor prefixes, and their removal leads to a reduction in the model’s classification effectiveness.

The analysis of ScikitRNN after removing individual protein descriptors revealed both positive and negative effects in the ScikitRNN_CountVectorizer model. For instance, when the CTDD_ScikitRNN_CountVectorizer combination had its descriptor prefix removed, several performance metrics improved significantly, particularly in Precision, Accuracy, and classification balance (Figure 5a, Appendix A). This indicates that the removal of the protein descriptor prefix optimized the overall classification performance of the model, especially in terms of balancing the differentiation between positive and negative samples.

When comparing the difference in performance metrics between models using full descriptors and those after prefix removal, the CTDD_ScikitRNN_CountVectorizer combination further demonstrated its superior performance and better balance in classifying positive and negative samples. Specifically, Precision increased by 0.502944, Accuracy improved by 0.550239, F1-Score rose by 0.371357, and MCC saw a significant increase of 0.920511 (Figure 5b, Appendix A).

Overall, it was observed that the convolutional neural network structure used by ScikitRNN generally exhibited weaker performance in terms of prediction capabilities, with only CTDD yielding favorable results. In other cases, the model exhibited clear disadvantages and lacked stability.

The analysis of protein descriptors across different models revealed that removing certain descriptors resulted in either positive or negative changes in performance, depending on the model structure and descriptor characteristics. Not all descriptors exhibited consistent effects across all models. In some cases, removing specific descriptors enhanced the model’s Accuracy, Recall, and F1-Score, while in other cases, performance worsened. Although some descriptors showed similar effects across different models, their overall impact on performance was typically small, as model performance is a comprehensive result influenced by numerous features. By systematically removing individual protein descriptors and monitoring the resulting performance changes, it is possible to determine model preferences for specific descriptors. This enables the selection of optimal descriptor combinations to improve model prediction capability and stability. This approach not only helps in understanding the importance of features but also provides a systematic method to screen key descriptors to enhance model accuracy and robustness.

To validate the feature-selection strategy, we identified relatively negative-impact protein descriptors for removal. In RandomForest_Doc2Vec, descriptors like AAC and GAAC displayed relatively weak consistency and were considered for removal in comparative tests. Similarly, in ScikitRNN_CountVectorizer, CTDD and DPC exhibited weaker consistency and were included in the comparison. By calculating the results after removing these paired descriptors and comparing them with the baseline and models with individual descriptors removed, we focused on analyzing changes in classification balance and classification accuracy.

Figure 6a (Appendix A) presents the results obtained after individually removing the AAC and GAAC feature sets. Although the Precision of the RandomForest model slightly decreased to 0.909871245, the Recall and Specificity remained at high levels, indicating that the model’s Precision and classification balance were not significantly affected. The model continued to effectively identify positive samples. Specifically, in the case of removing the AAC feature set, Precision reached 0.9375, MCC remained at 0.835436435, and Specificity was as high as 0.963350785, highlighting the model’s excellent ability to accurately identify negative samples. These results suggest that after removing AAC, the model’s performance in classifying negative samples significantly improved, demonstrating enhanced capability in negative sample screening. When no prefixes were removed, the RandomForest model maintained a Recall of 0.734694 and an Accuracy of 0.880383, showing relatively stable performance. However, compared to the combinations with individual prefix removal, Accuracy was slightly lower, indicating that the model’s classification accuracy was somewhat less optimal when no features were removed. This result demonstrates that the AAC_RandomForest_Doc2Vec combination, where AAC was removed, was the best-performing model in this study. Additionally, in tasks requiring a balance between positive and negative sample classification, removing AAC contributed to optimizing model performance.

In contrast, as shown in Figure 6b (Appendix A), after removing both CTDD and DPC features in the ScikitRNN_CountVectorizer model, the Recall dropped to 0.951020408, and the Precision was 0.872659176, resulting in overall underperformance. When only the CTDD feature was removed, the model’s Recall improved to 0.963265306, and both the F1-Score and Specificity reached 0.92, suggesting a more balanced classification of positive and negative samples. However, the overall performance did not surpass that of the combination where only CTDD was removed. By comparison, the combination that removed only the CTDD feature achieved the best performance, with a Recall of 0.963265306, Precision of 0.890566038, Specificity of 0.92408377, and MCC of 0.876459803, demonstrating strong classification balance and the ability to identify negative samples accurately. Thus, removing the CTDD feature proved to be the optimal choice for enhancing model performance.

In summary, selectively removing certain features, such as CTDD and DPC, can improve model performance, while excessive removal, such as eliminating both AAC and GAAC simultaneously, or not removing any features, can potentially degrade model performance. This underscores the importance of careful feature selection to avoid removing too many features at once. Although removing AAC and GAAC resulted in improvements in certain metrics, these feature sets encompass a large number of sub-features, and their removal reduced dimensionality, thereby affecting overall performance.

### 2.5. Selection of Optimal Machine Learning Models

Combining each machine learning model with its most compatible string feature-engineering method and protein descriptor set enables a more comprehensive performance evaluation. This approach allows for a deeper analysis of the overall advantages of different model-feature combinations, facilitating a more nuanced comparison. Among the top-performing model combinations, GradientBoosting_HashingVectorizer achieved the best Precision at 0.974025974025974, indicating that this model almost entirely eliminates false positives while identifying positive samples (Figure 7, Appendix A). Its Specificity reached 0.984293193717277, meaning that the model has extremely high accuracy in classifying negative samples, with almost no misclassification of negatives as positives. Additionally, the AUROC and AUPRC for this combination were the highest among all tested models, which is particularly important for the imbalanced sample test used in this study. Therefore, this model is highly suited for tasks that require high accuracy in identifying positive samples while strictly controlling for false positives in negative samples.

When optimizing models, it is crucial to avoid removing features that could decrease model performance consistency. If the removal of features does not lead to a consistent performance improvement, the best-performing feature combinations from the analysis should be prioritized. Model stability and overall performance should be key considerations in this selection process. Overall, GradientBoosting demonstrated balanced and stable performance across multiple evaluation metrics, indicating that it is less sensitive to different descriptor features and can effectively avoid over-reliance on local features. In contrast, although the ScikitRNN model showed strong performance in certain tasks, it exhibited higher sensitivity to specific features, leading to significant fluctuations in some cases. This over-sensitivity to local features may result in less stable accuracy in practical applications.

Therefore, in real-world scenarios, model selection should not only consider task-specific performance but also emphasize robustness to feature variations and adaptability to different situations. The GradientBoosting model was validated as being highly stable, and unlike other models, it does not overly rely on a single feature set. It demonstrates a strong overall perception of protein descriptors, making it a solid base model for further analysis. Specifically, during feature selection, GradientBoosting displayed high stability and balanced performance; even when some features were removed, its performance remained relatively stable without significant fluctuations or excessive dependence on a particular type of feature. On the other hand, models that excessively depend on local features may face performance instability in real-world applications.

### 2.6. Standardization and Oversampling of Models

Since data preprocessing methods and oversampling techniques can change the amount, shape, and distribution of the data, a further analysis of these methods was conducted based on the selected baseline model, GradientBoosting. This analysis focused on the impact of various preprocessing and oversampling techniques on model performance, particularly in addressing the issues of imbalanced data distribution and dispersed feature value ranges. The extracted feature values varied significantly, with some features ranging from 0 to 1, while others spanned from −8 to 123. These variations in feature value distributions directly influence the convergence speed and effectiveness of machine learning models. Additionally, the imbalance in the number of positive and negative samples in the training set can also negatively affect model performance. To address these issues, this study employed Min–Max Scaling (MinMaxScaler, python), Standardization (StandardScaler, python), and oversampling techniques such as SMOTE (Synthetic Minority Over-sampling Technique) and a cost-sensitive algorithm to preprocess the data. The effects of different preprocessing methods were then compared in detail.

As shown in Figure 8 (Appendix A), different preprocessing methods and oversampling techniques have a significant impact on model performance, with the SMOTE technique demonstrating particularly strong enhancement effects. It is evident that the model performed exceptionally well after MinMaxScaler normalization. Specifically, the model achieved a Recall of 0.922449 and a Precision of 0.961702, demonstrating strong positive-sample recognition capabilities. The overall Accuracy was 0.955343, with an F1-Score of 0.941667 and a Matthews Correlation Coefficient (MCC) of 0.906032, indicating good balance in classifying both positive and negative samples. Additionally, the model’s Area Under the Receiver Operating Characteristic Curve (AUROC) was 0.982178, further validating its robustness. In contrast, using StandardScaler for normalization resulted in a Precision of 0.985075, meaning almost no false positives were detected. However, the Recall was only 0.808163, indicating that a large number of positive samples were not correctly identified. Although this combination maintained high AUROC and Area Under the Precision–Recall Curve (AUPRC) scores, the low Recall made this model unsuitable as the best choice. In contrast, the cost-sensitive algorithm performs noticeably worse than other algorithms across most metrics. This approach may be more suitable for scenarios where there is a significant imbalance between positive and negative samples.

When SMOTE was applied for oversampling the minority class, the model’s performance improved significantly. Recall and Precision reached 0.922449 and 0.961702, respectively, and the F1-Score was up to 0.941667, indicating that the model achieved an ideal balance between Precision and Recall. MCC also improved to 0.906032, further confirming the model’s strong performance in classifying both positive and negative samples. The effect of SMOTE was particularly pronounced in addressing data imbalance. By increasing the number of minority-class samples, SMOTE effectively mitigated the negative impact of class imbalance on model performance, enabling the model to better recognize positive samples.

Overall, the combination of SMOTE and GradientBoosting_HashingVectorizer delivered the best performance. This model excelled in several key metrics, including Recall, Precision, F1-Score, and MCC, particularly in maintaining a balanced classification of positive and negative samples. It is highly suitable for scenarios that require both high Precision and high Recall. In comparison, both SMOTE and MinMaxScaler significantly improved model classification performance, particularly in balancing Recall and Precision. SMOTE further enhanced the Matthews Correlation Coefficient (MCC) and F1-Score, highlighting its effectiveness in addressing data imbalance issues. This comparative analysis underscores the importance of appropriate preprocessing and oversampling techniques for optimizing model performance, especially when working with complex and imbalanced datasets.

### 2.7. Cross-Validation Analysis

We compared the impact of varying cross-validation strategies on model performance, with a focus on the model’s ability to maintain balance and discrimination between positive and negative samples. We evaluated the selected baseline model using 10-fold, 2-fold, and 5-fold cross-validation to determine the optimal validation method for model performance.

First, using 10-fold cross-validation, the model achieved an Accuracy of 0.889952 and a Precision of 0.79932. While these metrics were slightly lower compared to the results from 2-fold and 5-fold cross-validation, the overall model performance remained strong. With 2-fold cross-validation, the Specificity improved, and both Precision and Accuracy slightly surpassed those found using the 10-fold validation. The results from the 5-fold cross-validation were generally consistent with those from the 2-fold validation, with variations in specific metrics showing slight differences. However, overall, the 5-fold cross-validation demonstrated superior performance (Figure 9, Appendix A).

### 2.8. Analyzing the Impact of Feature Selection on Models

In Figure 10 (Appendix A), key metrics such as Recall, Precision, F1-Score, and MCC are used to compare the performance of four feature-selection methods—Chi2, f_classif, Mutual Information, and Variance Threshold—focusing on their effectiveness in handling imbalanced data. The results indicate that the f_classif method performed exceptionally well across multiple evaluation metrics, particularly in terms of Recall and Precision, demonstrating a high degree of balance in distinguishing between positive and negative samples. The Recall for f_classif also remained at high levels, indicating outstanding overall classification performance. Moreover, the AUROC and AUPRC for the f_classif method were strong, further validating its powerful classification capabilities in handling complex datasets. Given its balanced performance across several key metrics, f_classif emerges as the best feature-selection method for the current dataset.

In comparison, the performance of the Chi2 method was slightly weaker than that of f_classif. While it maintained a high Precision, its lower Recall led to a reduced F1-Score. MCC also decreased significantly, suggesting that Chi2 struggles with the balanced classification of positive and negative samples. Although the method showed strong Specificity, it did not match the exceptional performance of f_classif in handling imbalanced data. The performance of the Mutual Information method was quite similar to f_classif. The Mutual Information method displayed balanced Accuracy and Precision, but its Recall was somewhat lower than that of f_classif. The performance of the variance threshold method was very similar to that of Chi2, but overall, both methods underperformed compared to f-classif.

Overall, the analysis of various metrics shows that the f_classif method holds a slight advantage, particularly in critical metrics such as Recall and AUROC. Thus, the f_classif method is the most suitable choice for feature selection and model construction in this dataset, particularly for complex tasks involving imbalanced distributions of positive and negative samples.

### 2.9. Performance of the TPGPred Model on Actual Thermophilic Proteins

Table 3 presents the results of testing thermophilic proteins referenced in the literature. These strains are distinct from those used in the training and test sets, each possessing specific functions and applications. However, they share similar thermophilic and non-thermophilic characteristics with the strains used in the training and test sets. The amino acid sequences of these proteins were downloaded from the NCBI database, with corresponding parameters and identifiers also sourced from NCBI. These data were not included in the model’s training or validation phases and were used solely for independent testing at the final stage. The selected model was applied to predict whether these proteins are thermophilic. Although the proteins selected for this test have already been identified as thermophilic, this does not imply that the model has a general capability to identify all thermophilic proteins. Due to the limited data available, both the model’s training and prediction capabilities are constrained. This is reflected in the performance evaluation results, which highlight the need for collecting more data in the future to further improve the model’s accuracy and robustness. The performance of the TPGPred model on actual thermophilic proteins demonstrates the model’s current limitations and potential for future improvement as additional data are collected.

### 2.10. Feature Importance Analysis

In Figure 11 (Appendix A), the importance of various descriptors in the machine learning model is analyzed, focusing on the cumulative effect of feature weights and their roles in model predictions. The weight of each feature is the sum of the weights of all its internal sub-features. Due to the varying number and impact of sub-features within each descriptor, a positive cumulative weight does not necessarily mean that all sub-features contribute positively to the model. By comparing these weights, we can identify which features are most critical to model predictions and which have a lesser impact.

Key features were analyzed individually. The cumulative weight of the str_feature is 0.48736, making it the highest-weighted feature, indicating that this feature (likely representing direct protein-sequence encoding) has the most significant influence on the model, far surpassing other features. This feature is derived from string-based feature engineering, reflecting the global encoding of amino acid sequences. Such global sequence features play a crucial role in recognizing overall protein information. The GAAC (Global Amino Acid Composition) feature ranks second, with a cumulative weight of 0.23415. This feature reflects the overall composition of amino acids, indicating that the global proportion of amino acids has a substantial impact on model predictions. This suggests that the composition of protein sequences is one of the key factors driving the model’s decisions. The AAC (Amino Acid Composition) feature ranks third, with a cumulative weight of 0.08287. Although its influence is less than GAAC, it still makes a significant contribution to the model, showing that the specific types and quantities of individual amino acids affect predictions. The cumulative weight of DDE (Dipeptide Deviation from Expected mean) is 0.05283, indicating that dipeptide combinations contribute to model predictions, though their impact is lower compared to global amino acid composition features. This feature reflects local sequence information and its influence on the model. The cumulative weights of TPC (Tripeptide Composition) and DPC (Dipeptide Composition) are 0.036106 and 0.014825, respectively. These features capture local sequence segment information, but their importance is lower compared to global features like GAAC and AAC. Lower-weighted features such as CKSAAGP, CTDC, CTDD, CTDT, and CTriad have cumulative weights ranging from 0.001475 to 0.036844. These are based on more complex or sparse descriptor combinations and may not capture information that significantly aids in model decisions.

As shown in Figure 12 (Appendix A), SHAP analysis [40,41] was used to rank and extract the contributions of protein descriptors based on feature weights, displaying the top 30 sub-features with the highest importance. Among these, several sub-features of the str_feature descriptor were ranked at the top, with feature_599 having the highest weight at 0.302314, far exceeding other features. This highlights the critical role of string-based features in the model. The uncharge sub-feature from GAAC (Global Amino Acid Composition) had a weight of 0.23415, ranking second among the features, indicating that the global composition of uncharged amino acids significantly influences the model’s predictions. From a biological and structural perspective, uncharged amino acids play a critical role in forming the hydrophobic core of proteins. The proportion of uncharged residues is likely associated with protein stability.

Several sub-features from AAC (Amino Acid Composition) also ranked highly, such as amino acids P (0.041766), I (0.025993), D (0.003952), E (0.003506), and H (0.002281). These findings imply that specific amino acids, particularly P and I, play critical roles in the model, potentially due to their strong associations with protein function or structure. Additionally, sub-features from DDE (Dipeptide Deviation from Expected mean), such as QL.1 (0.007311), QQ.1 (0.008557), and EQ.1 (0.015675), also appeared in the top rankings. Although these dipeptide combinations had lower weights, they still contributed to the model, indicating that local amino acid sequence patterns play a supporting role in predictions. A sub-feature from CTDC (Composition, Transition, and Distribution of various physicochemical properties), hydrophobicity_CASG920101.G3, had a weight of 0.017529, suggesting that physicochemical properties related to hydrophobicity have some impact on model predictions. These features may relate to the physicochemical stability or structural attributes of proteins and thus have a secondary influence on prediction outcomes. Sub-features from DPC (Dipeptide Composition) and TPC (Tripeptide Composition), such as PE, QL, and KDL, had weights ranging from 0.001864 to 0.005718. Although lower in importance, these features still contributed by capturing short peptide sequence patterns in proteins, playing a minor but notable role in the model. From a biological perspective, the hydrophobic structure of proteins plays an important role in the identification of thermophilic proteins. For example, uncharged amino acids (e.g., glycine and alanine) in the GAAC descriptor, isoleucine (I) in the AAC descriptor; dipeptide patterns such as QL (glutamine-leucine) and EQ (glutamic acid-glutamine) in the DDE (Dipeptide Deviation from Expected mean) descriptor; and the “hydrophobicity_CASG920101.G3” feature in the CTDC descriptor, all related to hydrophobicity, contribute significantly to the model (I) in the AAC descript.

## 3. Discussion

Overall, global features (such as str_feature and GAAC) are much more important than local features (such as TPC and DPC). They represent global information about protein sequences and amino acid composition. The model relies more on overall sequence characteristics and amino acid ratios rather than localized segment information. The high weight of global features suggests that the model considers the overall composition and structure of the sequence as critical for accurate predictions. The high importance of GAAC and AAC implies that the types and proportions of amino acids play a crucial role in predictions, indicating that the model relies more on global amino acid composition rather than their specific positions or local sequence order. Local features like DDE, DPC, and TPC, while contributing to some extent, are clearly less influential than global features. This suggests that although local features reflect some local patterns, their impact on overall classification is relatively limited. Complex or sparse features such as CKSAAGP and KSCTriad contribute very little to the model, possibly due to their inability to effectively capture key information from protein sequences, or because these features are too sparse to be influential in the model. The slight performance improvement after removing the GAAC feature in the baseline model can be explained by the feature importance analysis. The higher global representation power of str_feature compared to GAAC means that removing GAAC slightly strengthens the weight of str_feature, leading to a marginal impact on overall results, though the effect remains minimal.

String-based features were shown to be highly significant, with sub-features from str_feature dominating the importance rankings, particularly feature_599, whose weight far surpassed that of other features. Amino acid composition features from GAAC and AAC were also notable, with high-weight sub-features focusing on the charge state of amino acids (e.g., uncharged amino acids) and specific amino acid types (e.g., P, I, C). The high importance of these amino acid composition features suggests that the types and proportions of amino acids are crucial in predicting protein function, with amino acids P and I potentially playing a particularly influential role in the model. While local sequence features such as DDE, DPC, and TPC were less important than global features, they still made meaningful contributions to the model. Notably, certain dipeptide combinations from DDE (e.g., QL.1 and QQ.1) had relatively high weights, indicating that these local sequence patterns may be linked to specific protein functions. Physicochemical properties, such as hydrophobicity (as reflected by CTDC), also contributed to the model, likely due to their relationship with protein structural stability or function. Also, the model relies heavily on global features like str_feature and amino acid composition (GAAC, AAC) for predictions, while local sequence patterns and physicochemical properties play supplementary roles. The findings suggest that the types and global distribution of amino acids are central to protein function prediction, and features capturing global sequence characteristics are particularly valuable.

## 4. Materials and Methods

### 4.1. Construction of Thermophilic and Non-Thermophilic Protein Datasets

We downloaded all thermophilic protein sequences of *Thermus aquaticus* from the NCBI database [42] to construct the thermophilic protein dataset. *Halomonas* TD01 (Beijing, China) a chassis organism long-studied and engineered by our laboratory. Over years of wet-lab experiments, it has been proven to be non-thermophilic, and its complete-genome sequence is stored in our laboratory’s internal database. All amino acid sequences were analyzed, and sequences with missing values, non-amino acid characters, or abnormal structures were excluded. Pairing was conducted using a basic amino acid element list, and non-conforming pairs were removed. Regarding the independent test set, using keywords related to thermophilic proteins, we conducted a literature search and downloaded relevant protein sequences from specific strains in the NCBI database. These strains exhibit extreme thermophilic characteristics. The protein sequences in the independent test set underwent the same data processing as the training and validation sets, including checks for invalid sequences, the removal of outliers, and feature construction.

### 4.2. String Feature-Engineering Construction

The feature-engineering techniques used in this study involve two main approaches: the direct construction of sequence features from amino acid strings and protein-encoding-based features. The amino acid sequences were treated as ordinary text strings, and traditional text feature-extraction methods were employed [24,43], including grams, n-gram, hashing, LDA, LSA, PCA, T-SNE, Word2Vec, FastText, Doc2Vec and BERT. The grams and N-grams methods break down the text into character-level n-grams and convert the frequency of these n-grams into a feature matrix. The hashing method maps character-level n-grams to a fixed-size feature space using a hashing function, generating a sparse matrix without relying on a vocabulary, which is memory-efficient for large-scale text processing. Latent Semantic Analysis (LSA) reduces high-dimensional representations to a low-dimensional semantic space through singular value decomposition (SVD). Latent Dirichlet Allocation (LDA) is a probabilistic topic modeling method often used in biological literature analysis. Principal Component Analysis (PCA) is a linear dimensionality reduction technique widely used for gene expression analysis; it reveals key patterns and differences in biological data, making it useful for identifying major trends in protein functionality and gene expression. T-SNE is a non-linear dimensionality reduction method for visualizing high-dimensional data. Word2Vec is a neural network-based word embedding model that captures semantic relationships between biological terms, such as genes or proteins. FastText is an extension of Word2Vec that improves handling of spelling variations by training on sub-word embeddings. Doc2Vec is an extension of Word2Vec that represents entire documents as vectors. It is useful for biological literature classification, clustering, and retrieval, enabling the identification of functionally related documents by encoding global semantic information from texts like genome annotations or abstracts. BERT uses transformer embedding to extract string features of protein sequences.

### 4.3. Feature Descriptors

In the experiment, 14 feature descriptors were used [44,45,46]. Each feature captures distinct aspects of the protein sequences. AAC (Amino Acid Composition) refers to the frequency of each amino acid in the protein sequence. CKSAAP (Composition of K-Spaced Amino Acid Pairs) refers to the frequency of amino acid pairs separated by a specific interval, K. CTDC (Composition of Physicochemical Groups) describes the composition of the protein sequence based on the physicochemical properties of amino acids. CTDT (Transition of Physicochemical Groups) refers to the transition frequencies between amino acids belonging to different physicochemical groups. CTDD (Distribution of Physicochemical Groups) describes the distribution patterns of each physicochemical group in the amino acid sequence. CTriad (Composition of Amino Acid Triplets Based on Physicochemical Properties) refers to the frequency of amino acid triplets (trinucleotides) based on their physicochemical properties. DPC (Dipeptide Composition) describes the frequency of consecutive dipeptides in the protein sequence. GAAC (Grouped Amino Acid Composition) refers to the frequency of amino acids grouped by predefined categories. GDPC (Grouped Dipeptide Composition) describes the frequency of consecutive dipeptides based on predefined amino acid groups. CKSAAGP (Composition of K-Spaced Amino Acid Group Pairs) refers to the frequency of pairs of amino acid groups separated by a specific interval, K. DDE (Dipeptide Deviation from Expected Frequency) refers to the deviation between the observed and expected frequency of dipeptides. GTPC (Grouped Tripeptide Composition) describes the frequency of tripeptides based on predefined amino acid groups. KSCTriad (K-Spaced Triad Composition) refers to the frequency of amino acid triads separated by an interval, K. TPC (Tripeptide Composition) refers to the frequency of tripeptides in the sequence.

These feature descriptors extract key information from protein sequences, such as amino acid composition, physicochemical properties, and local sequence patterns, for protein function classification. Each descriptor has its unique advantages in capturing different patterns and regularities within protein sequences, making them useful for bioinformatics-based machine learning and deep learning models.

### 4.4. Feature Standardization

Min–Max Normalization [47] was applied to scale the data to the range of [0, 1], eliminating differences in scale between features and improving model performance. This linear transformation preserves the relative size of the data but is sensitive to outliers.
(1)$$x_{i}^{\prime }=\frac{x_{i}-x_{min}}{x_{max}-x_{min}}$$
where $x_{i}$ is the original data point, $x_{min}$ and $x_{max}$ are the minimum and maximum values of the feature, and $x_{i}^{\prime }$ is the normalized value.

Z-Score Standardization [48] transforms the data so that the mean is 0 and the standard deviation is 1. This method eliminates differences in scale between features, ensuring they share the same distribution characteristics. It is particularly suitable for most machine learning models, especially those that rely on distance-based metrics.
(2)$$z_{i}=\frac{x_{i}-\frac{1}{N}\sum_{i=1}^{N}x_{i}}{\sigma \sqrt{\frac{1}{N}\sum_{i=1}^{N}\left(x_{i}-\frac{1}{N}\sum_{i=1}^{N}x_{i}\right)}}$$
where $z_{i}$ is the standardized data point. After this transformation, the resulting distribution of $z_{i}$ has a mean of 0 and a standard deviation of 1.

SMOTE [49] is an oversampling technique designed to address class imbalance by synthesizing new minority-class samples to balance the class distribution. It achieves this by performing linear interpolation between existing minority-class samples in feature space, enhancing the representation of the minority class and improving model training. The main steps include the following: (1) Randomly selecting a minority-class sample and its nearest neighbor. (2) Generating new synthetic samples by performing linear interpolation between the selected sample and its neighbor. SMOTE helps improve the model’s ability to recognize minority-class samples and is particularly effective for dealing with severely imbalanced datasets.

### 4.5. Feature Selection

Chi2 (Chi-Square Test) [50] is a feature-selection method based on the chi-square test, used to measure the independence between discrete features and the target variable, and it is commonly applied in classification tasks. For each feature, the Chi2 method assesses the relationship between the feature and the target variable by calculating the difference between observed and expected values. A larger difference indicates a stronger correlation between the feature and the target variable.
(3)$$\chi^{2}=\sum \frac{{\left({ο}_{i}-E_{i}\right)}^{2}}{E_{i}}$$
f_classif [51] uses ANOVA (Analysis of Variance) to evaluate the relationship between each feature and the target variable, primarily applied in classification tasks. It assesses this relationship by calculating the ratio of variance between groups to the variance within groups. The significance of each feature is measured based on the F-statistic, which determines whether the variance explained by the feature is significantly greater than the variance within the target variable groups.
(4)$$F=\frac{Mean\;Squared\;Between}{Mean\;Squared\;Within}$$

Mutual Information [52] is used to measure the mutual dependence (or information gain) between a feature and the target variable, capturing nonlinear relationships. Mutual information quantifies how much information a feature provides about the target variable. It is calculated by evaluating the difference between the joint probability distribution of the feature and the target variable and their marginal probability distributions. This difference reflects the contribution of the feature to the target variable’s prediction.
(5)$$I(X;Y)=\sum_{x\in X}\sum_{y\in Y}P(x,y)log(\frac{P(x,y)}{P(x)P(y)})$$
where P(x,y) is the joint probability distribution, and P(x) and P(y) are the marginal probability distributions of the feature and the target variable, respectively.

Variance Threshold [53] is a simple feature-selection method that removes features whose variance falls below a specified threshold, primarily aimed at reducing data redundancy. Features with low variance exhibit little variation across the dataset and may not contain useful information.
(6)$$Var(X)=\frac{1}{n}\sum_{i=1}^{n}{\left(x_{i}-\mu \right)}^{2}$$
where μ is the mean of the feature, and $x_{i}$ denotes the feature values.

### 4.6. Random Ten-Fold Cross-Validation

Common cross-validation methods include two-fold, five-fold, and ten-fold cross-validation. In cross-validation, the dataset is divided into several parts (folds) [54]. In each iteration, one fold is used as the validation set while the remaining folds are used for training. This process is repeated so that every fold is used for both training and validation, ensuring that all data are involved in the model evaluation. Cross-validation effectively assesses the generalization performance of the model and reduces bias from single data splits, making it suitable for various machine learning tasks. Two-fold cross-validation is split into two parts—one for training and the other for validation. This process is repeated twice to ensure all data are used for both training and validation. Five-fold cross-validation is divided into five parts. In each iteration, one part is used for validation, and the rest are used for training. This is repeated five times. Ten-fold cross-validation is divided into ten parts. In each iteration, one part is used for validation, and the remaining parts are used for training. This process is repeated ten times. It is ideal for larger datasets, offering comprehensive and stable performance evaluations at a higher computational cost.

### 4.7. Performance Evaluation

Eight commonly used metrics were employed to evaluate and compare the model’s performance: Recall, Precision, Accuracy, F1-Score, MCC, Specificity, AUROC, and AUPRC [55,56]. The specific calculations are as follows:

Recall measures the model’s ability to correctly identify all positive samples.
(7)$$Recall=\frac{TP}{TP+FN}$$

Precision evaluates the proportion of correctly predicted positive samples among all samples.
(8)$$Precision=\frac{TP}{TP+FP}$$

Accuracy represents the proportion of correctly predicted samples out of all samples.
(9)$$Accuracy=\frac{TP+TN}{TP+TP+FP+FN}$$

F1-Score is the harmonic mean of Precision and Recall, providing a balanced assessment of both.
(10)$$F1=2\frac{Precision\cdot Recall}{Precision+Recall}$$

MCC (Matthews Correlation Coefficient) is a metric that considers true positives, false positives, true negatives, and false negatives to measure the overall performance of the classification model.
(11)$$MCC=\frac{TP\cdot TN-FP\cdot FN}{\sqrt{\left(TP+FP\right)\left(FP+FN\right)\left(TN+FP\right)(TN+FN)}}$$

Specificity represents the proportion of correctly identified negative samples, assessing the model’s ability to recognize negative samples.
(12)$$Specificity=\frac{TN}{TN+FP}$$

AUROC (Area Under the Receiver Operating Characteristic Curve) evaluates the model’s ability to distinguish between positive and negative samples. The larger the area under the curve, the better the model’s classification performance.

AUPRC (Area Under the Precision–Recall Curve) measures the model’s Precision at different Recall levels; it is particularly useful for evaluating models based on imbalanced datasets by focusing on the performance on the minority class.

TP (True Positive): The number of samples that are actually positive and correctly predicted as positive.

TN (True Negative)**:** The number of samples that are actually negative and correctly predicted as negative.

FP (False Positive)**:** The number of samples that are actually negative but incorrectly predicted as positive.

FN (False Negative)**:** The number of samples that are actually positive but incorrectly predicted as negative.

### 4.8. Feature Importance Analysis

SHAP (SHapley Additive exPlanations) [40,41] is a model interpretation tool based on game theory, providing an intuitive way to explain machine learning models by quantifying feature contributions. SHAP values measure the positive or negative contribution of features to predictions, allowing for an assessment of both overall feature importance and the reasoning behind predictions for individual samples.

## 5. Conclusions

Thermophilic proteins, renowned for their high-temperature stability and unique structural properties, are invaluable in bioengineering and industrial applications. However, their experimental validation is complex and costly, necessitating efficient bioinformatics tools. Here, we introduce TPGPred, a model that rapidly identifies thermophilic proteins by integrating multiple feature-engineering methods and machine learning algorithms. Utilizing a comprehensive dataset, the GradientBoosting model combined with HashingVectorizer achieved superior Accuracy, Recall, and F1-Score values. Feature-elimination experiments revealed that global features significantly enhance prediction accuracy. TPGPred demonstrates strong predictive capability, highlighting its potential for practical applications in thermophilic protein identification and functional research.

## Figures and Tables

**Figure 1 ijms-25-11866-f001:**
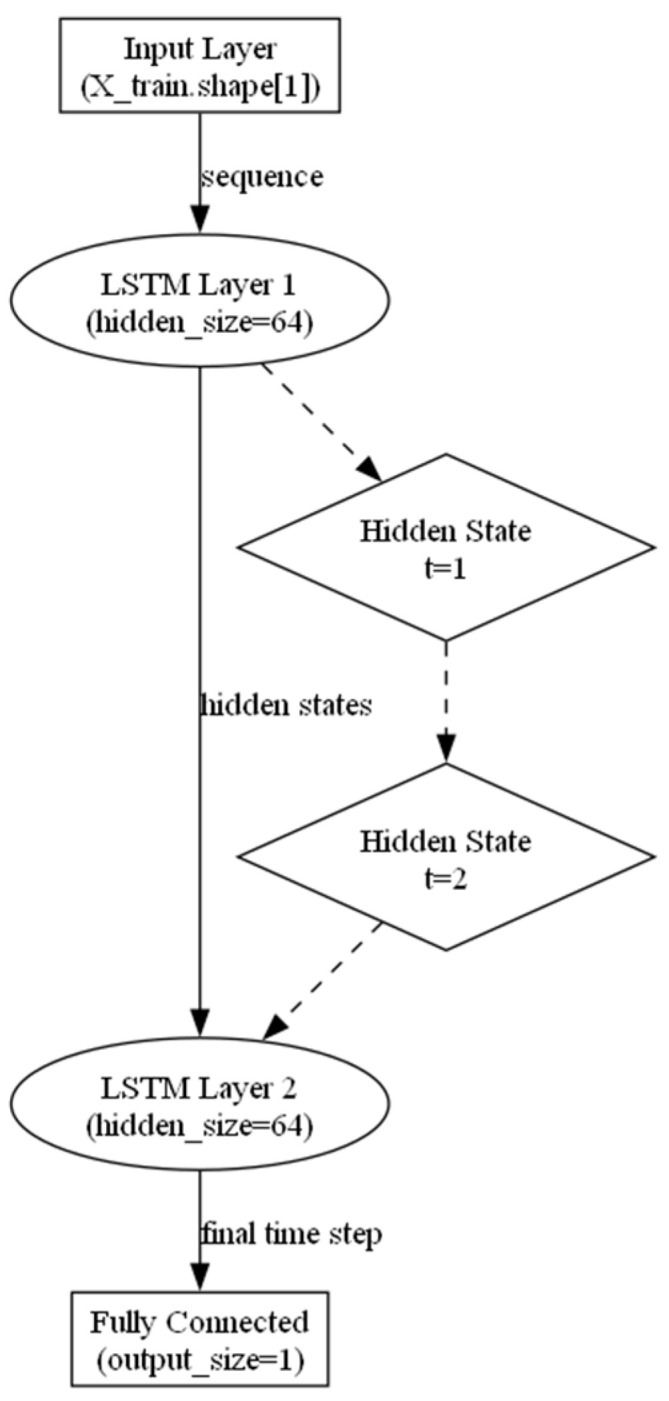
ScikitRNN network architecture design.

**Figure 2 ijms-25-11866-f002:**
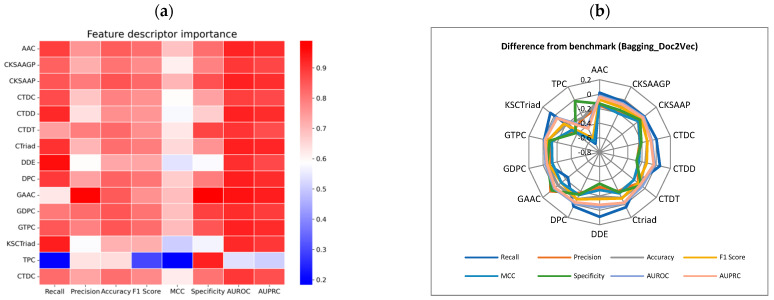
Performance analysis of the Bagging_Doc2Vec model after selecting protein descriptors. (**a**) Performance analysis of the Bagging_Doc2Vec model after removing the corresponding protein descriptors. (**b**) Difference in performance metrics between the Bagging_Doc2Vec model with individual protein descriptors removed and the baseline results. Baseline metrics were calculated using the full set of protein descriptors.

**Figure 3 ijms-25-11866-f003:**
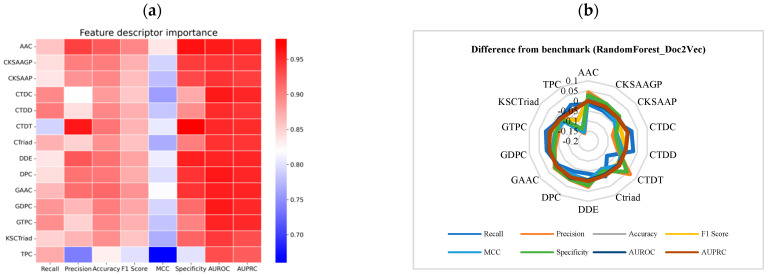
Performance analysis of the RandomForest_Doc2Vec model after selecting protein descriptors. (**a**) Performance analysis of the RandomForest_Doc2Vec model after removing the corresponding protein descriptors. (**b**) Difference in performance metrics between the RandomForest_Doc2Vec model with individual protein descriptors removed and the baseline results. Baseline metrics were calculated using the full set of protein descriptors.

**Figure 4 ijms-25-11866-f004:**
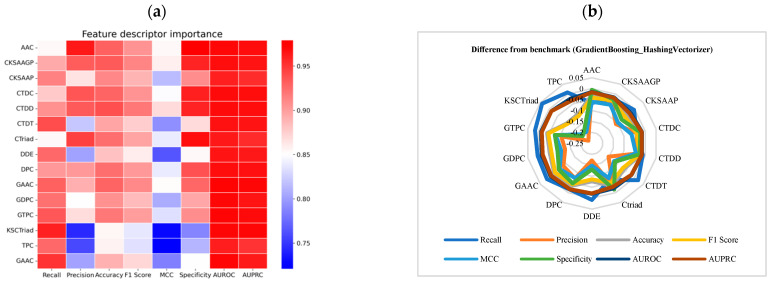
Performance analysis of the GradientBoosting_HashingVectorizer model after selecting protein descriptors. (**a**) Performance analysis of the GradientBoosting_HashingVectorizer model after removing the corresponding protein descriptors. (**b**) Difference in performance metrics between the GradientBoosting_HashingVectorizer model with individual protein descriptors removed and the baseline results. Baseline metrics were calculated using the full set of protein descriptors.

**Figure 5 ijms-25-11866-f005:**
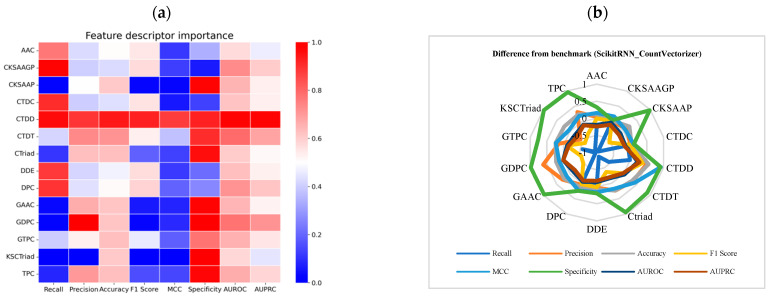
Performance analysis of the ScikitRNN_CountVectorizer model after selecting protein descriptors. (**a**) Performance analysis of the ScikitRNN_CountVectorizer model after removing the corresponding protein descriptors. (**b**) Difference in performance metrics between the ScikitRNN_CountVectorizer model with individual protein descriptors removed and the baseline results. Baseline metrics were calculated using the full set of protein descriptors.

**Figure 6 ijms-25-11866-f006:**
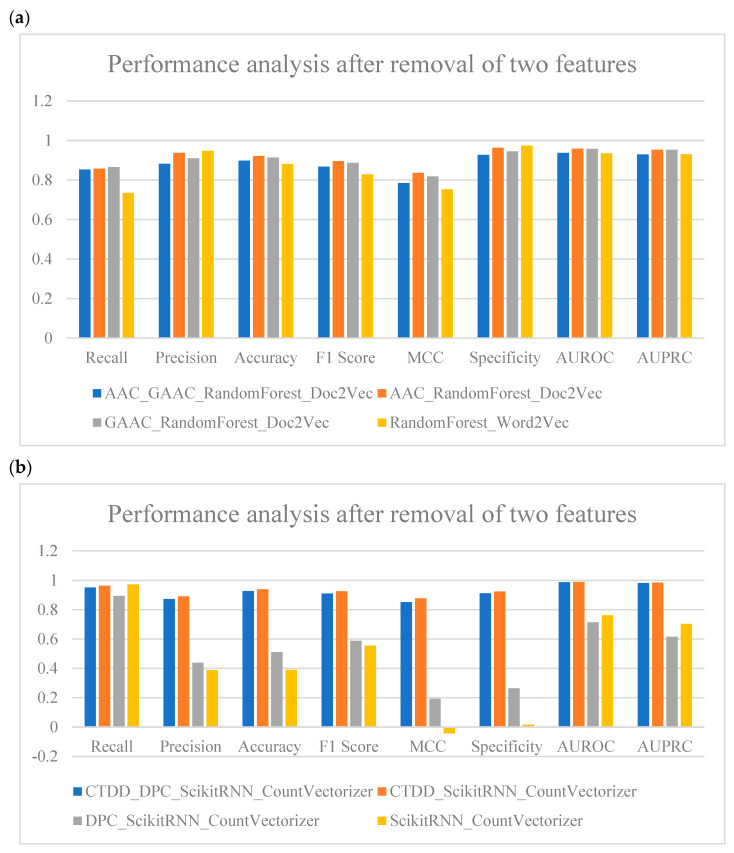
Comparison of model performance after simultaneous removal of paired protein description features. (**a**) Performance comparison of the RandomForest_Doc2Vec model after the simultaneous removal of AAC and GAAC protein descriptor features. (**b**) Performance comparison of the ScikitRNN_CountVectorizer model after the simultaneous removal of CTDD and DPC protein descriptor features.

**Figure 7 ijms-25-11866-f007:**
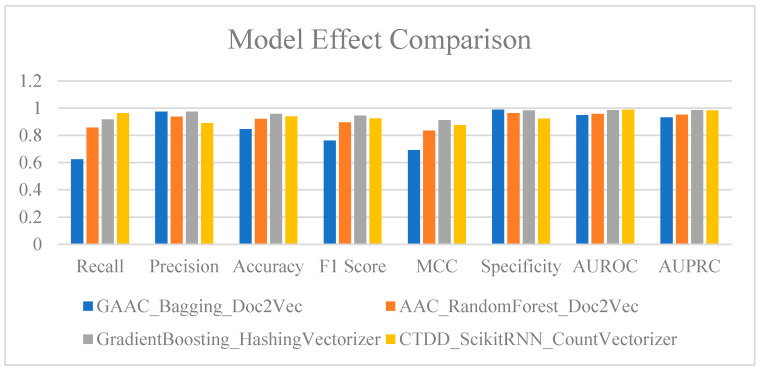
Performance comparison of different model series after applying various preprocessing techniques.

**Figure 8 ijms-25-11866-f008:**
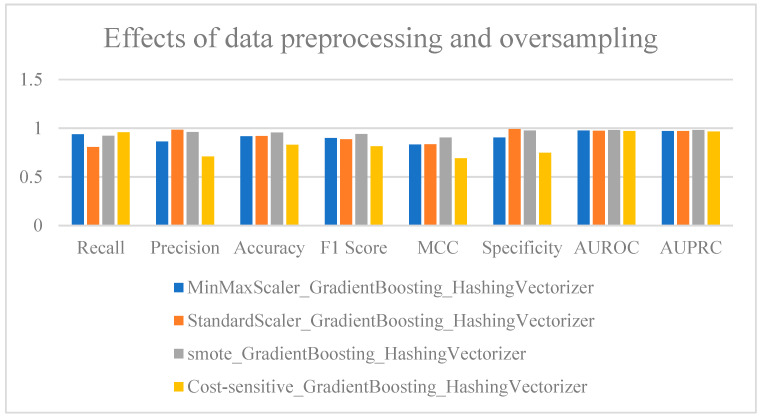
Comparison of results from data normalization and oversampling techniques.

**Figure 9 ijms-25-11866-f009:**
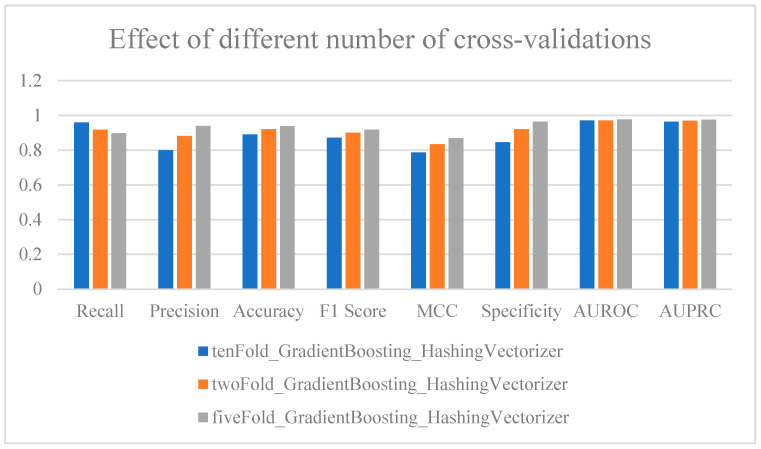
Analysis of the impact of different cross-validation methods on results.

**Figure 10 ijms-25-11866-f010:**
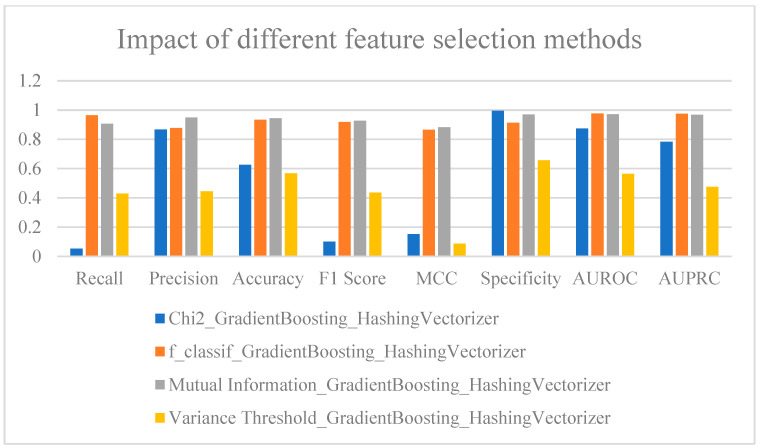
Analysis of the impact of different feature-selection methods on results.

**Figure 11 ijms-25-11866-f011:**
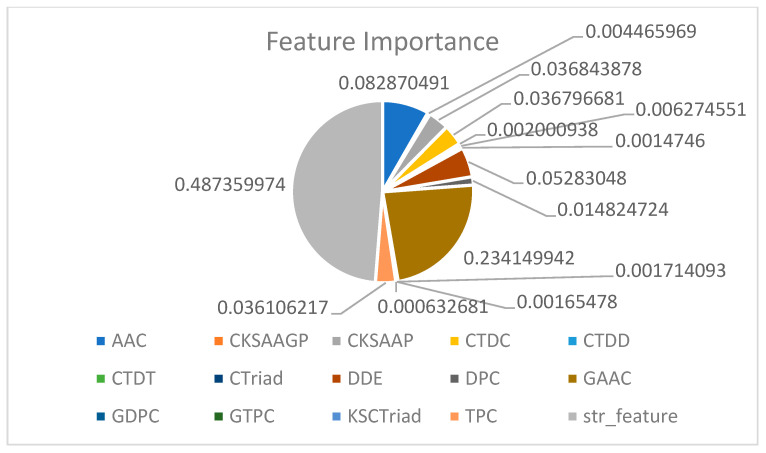
Cumulative analysis of feature weights for different descriptors.

**Figure 12 ijms-25-11866-f012:**
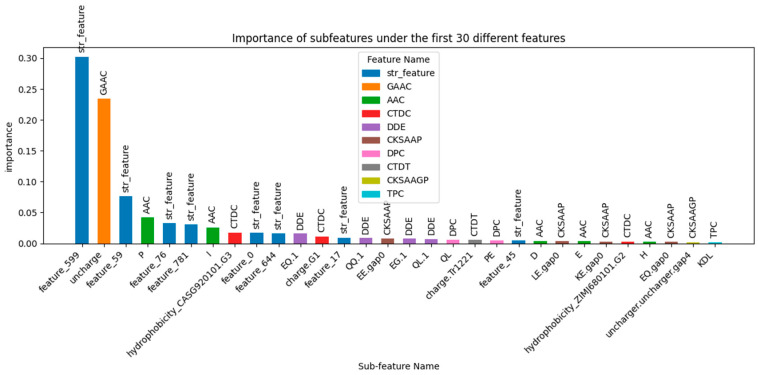
Importance ranking of the top 30 sub-features.

**Table 1 ijms-25-11866-t001:** Composition of the dataset.

Application	Protein Classification	Source	Name	Quantity
Training	Thermophilic protein	NCBI	*Thermus aquaticus*	2213
Training	Non-thermophilic protein	Sequencing in our laboratory and NCBI	*Halomonas* sp. TD01	3439
Testing	Thermophilic protein	NCBI	*Thermus aquaticus*	245
Testing	Non-thermophilic protein	NCBI	*Halomonas* sp. TD01	382

**Table 2 ijms-25-11866-t002:** Comparative analysis of different machine learning algorithms and corresponding string feature-engineering methods.

	Recall	Precision	Accuracy	F1-Score	MCC	Specificity	AUROC	AUPRC
Bagging_CountVectorizer	0.808163	0.838983	0.864434	0.823285	0.713715	0.900524	0.918549	0.89672
Bagging_CountVectorizer	0.808163	0.838983	0.864434	0.823285	0.713715	0.900524	0.918549	0.89672
Bagging_CountVectorizer_n_grams	0.906122	0.560606	0.685805	0.692668	0.455802	0.544503	0.881766	0.858188
Bagging_HashingVectorizer	0.893878	0.706452	0.813397	0.789189	0.639855	0.76178	0.931152	0.926136
Bagging_LDA	0.938776	0.651558	0.779904	0.769231	0.606718	0.67801	0.931141	0.91374
Bagging_LSA	0.816327	0.829876	0.862839	0.823045	0.711143	0.89267	0.927717	0.900711
Bagging_PCA	0.918367	0.650289	0.77512	0.761421	0.590257	0.683246	0.926648	0.913162
Bagging_t-SNE	0.767347	0.767347	0.818182	0.767347	0.618132	0.850785	0.884945	0.837632
Bagging_Word2Vec	0.869388	0.8875	0.905901	0.878351	0.801754	0.929319	0.962731	0.957469
Bagging_FastText	0.518367	0.686486	0.719298	0.590698	0.392133	0.848168	0.754541	0.710745
Bagging_Doc2Vec	0.865306	0.9098712	0.913875	0.8870292	0.8181855	0.9450261	0.9500641	0.9469430
Bagging_BERT	0.963265	0.482618	0.582137	0.643052	0.354431	0.337696	0.862159	0.834961
GradientBoosting_CountVectorizer	0.995918	0.458647	0.539075	0.628057	0.329304	0.246073	0.937974	0.913477
GradientBoosting_CountVectorizer_n_grams	1	0.492958	0.598086	0.660377	0.409586	0.340314	0.955786	0.94162
GradientBoosting_HashingVectorizer	0.918367	0.9740259	0.9585326	0.9453781	0.9130311	0.9842931	0.9851907	0.9846906
GradientBoosting_LDA	0.963265	0.670455	0.800638	0.79062	0.648572	0.696335	0.965028	0.957619
GradientBoosting_LSA	0.955102	0.735849	0.848485	0.831261	0.717516	0.780105	0.96584	0.954213
GradientBoosting_PCA	0.971429	0.636364	0.77193	0.768982	0.612042	0.643979	0.969366	0.962006
GradientBoosting_t-SNE	0.967347	0.642276	0.776715	0.771987	0.61651	0.65445	0.96848	0.964892
GradientBoosting_Word2Vec	0.877551	0.972851	0.942584	0.922747	0.880206	0.984293	0.977743	0.976425
GradientBoosting_FastText	0.918367	0.865385	0.912281	0.891089	0.818778	0.908377	0.966054	0.955491
GradientBoosting_Doc2Vec	0.946939	0.781145	0.875598	0.856089	0.759064	0.829843	0.968768	0.965108
GradientBoosting_BERT	1	0.427574	0.476874	0.599022	0.24585	0.141361	0.942141	0.924382
RandomForest_CountVectorizer	0.865306	0.625369	0.744817	0.726027	0.521699	0.667539	0.842366	0.766364
RandomForest_CountVectorizer_n_grams	0.897959	0.572917	0.698565	0.699523	0.469337	0.570681	0.852719	0.801474
RandomForest_HashingVectorizer	0.795918	0.709091	0.792663	0.75	0.576688	0.790576	0.870595	0.821572
RandomForest_LDA	0.844898	0.877119	0.893142	0.860707	0.774437	0.924084	0.939331	0.926095
RandomForest_LSA	0.934694	0.636111	0.76555	0.757025	0.583922	0.657068	0.930516	0.916145
RandomForest_PCA	0.767347	0.780083	0.824561	0.773663	0.630506	0.861257	0.890736	0.864668
RandomForest_t-SNE	0.832653	0.766917	0.835726	0.798434	0.661792	0.837696	0.906454	0.892166
RandomForest_Word2Vec	0.734694	0.947368	0.880383	0.827586	0.752223	0.973822	0.935271	0.930686
RandomForest_FastText	0.8	0.933333	0.899522	0.861538	0.789153	0.963351	0.951469	0.945652
RandomForest_Doc2Vec	0.873469	0.895397	0.910686	0.884298	0.811754	0.934555	0.958051	0.952242
RandomForest_BERT	0.906122	0.556391	0.681021	0.689441	0.449098	0.536649	0.84478	0.77856
ScikitRNN_CountVectorizer	0.971429	0.387622	0.389155	0.554133	−0.04405	0.015707	0.761064	0.703102
ScikitRNN_CountVectorize r_n_grams	0.995918	0.400657	0.416268	0.571429	0.118107	0.044503	0.726114	0.611873
ScikitRNN_HashingVectorizer	1	0.391374	0.392344	0.562572	0.032008	0.002618	0.752821	0.690301
ScikitRNN_LDA	0.979592	0.38961	0.392344	0.557491	−0.01747	0.015707	0.726157	0.653844
ScikitRNN_LSA	1	0.39075	0.39075	0.561927	0	0	0.611241	0.516229
ScikitRNN_PCA	1	0.39075	0.39075	0.561927	0	0	0.726979	0.653761
ScikitRNN_t-SNE	0.832653	0.392308	0.430622	0.533333	0.00704	0.172775	0.573234	0.452664
ScikitRNN_Word2Vec	0.910204	0.401802	0.435407	0.5575	0.06289	0.13089	0.632001	0.5229
ScikitRNN_FastText	0.946939	0.392555	0.406699	0.555024	0.014992	0.060209	0.61107	0.495664
ScikitRNN_Doc2Vec	0.995918	0.3904	0.39075	0.56092	−0.01267	0.002618	0.67785	0.589583
ScikitRNN_BERT	0.987755	0.391586	0.395534	0.560834	0.014201	0.015707	0.633289	0.506693

**Table 3 ijms-25-11866-t003:** Results of TPGPred testing on data introduced from the literature and feature importance analysis.

Source	LOCUS (Version)	True Type	Predicted Type	Reference
*Thermus thermophilus*	WP_143586044.1	TP	TP	[26]
*Thermus parvatiensis*	WP_008631403.1	TP	TP	[27]
*Thermus scotoductus*	WP_172960035.1	TP	TP	[28]
*Thermus composti*	WP_188845765.1	TP	TP	[29]
*Thermaceae*	WP_318773468.1	TP	TP	[30]
*Thermus islandicus*	WP_245540704.1	TP	TP	[31]
*Meiothermus* sp.	WP_314136165.1	TP	TP	[32]
*Vreelandella neptunia*	WP_133729827.1	NTP	NTP	[33]
*Pseudomonadota bacterium*	MEC9020573.1	NTP	NTP	[34]
*Gammaproteobacteria bacterium*	MBR9902729.1	NTP	NTP	[35]
*Halovibrio variabilis*	WP_146875438.1	NTP	NTP	[36]
*Natronocella acetinitrilica*	WP_253485102.1	NTP	NTP	[37]
*Actinomycetota bacterium*	MDQ2621912.1	NTP	NTP	[38]
*Microbacterium aquimaris*	WP_322602619.1	NTP	NTP	[39]

## Data Availability

Appendix A for this work can be found in an e-version of this paper online. Source code and curated datasets are publicly available at https://github.com/TPGPred/TPGPred (accessed on 10 October 2024).

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
