# Peer review of "TPGPred: A Mixed-Feature-Driven Approach for Identifying Thermophilic Proteins Based on GradientBoosting"

_ijms, 2024, doi:10.3390/ijms252211866_

Round 1
Reviewer 1 Report
Comments and Suggestions for Authors
The manuscript 'TPGPred: A Mixed-Feature-Driven Approach for Identifying Thermophilic Proteins Based on Gradient Boosting' presents a solid foundation for scientific research. However, in its current form, it may not meet the standards required for publication in IJMS. I recommend a major revision to address some key aspects of the methodology, clarity, and depth of analysis. After these improvements, the manuscript could be suitable for publication.
1. The test set appears to be disproportionately small compared to the training data, which may limit the robustness of the external validation. I recommend allocating at least 10% of the dataset for testing to ensure a more reliable and fair evaluation of the model's performance. Additionally, the test set should be stratified to ensure that the proportions of positive and negative class observations are consistent with those in the training data, which will help maintain representativeness and improve the validity of the results.
2. The authors have only considered two organisms for building their models, which raises questions about the generalizability of the results. I suggest the authors provide a more thorough justification for this choice in the manuscript. Additionally, a comprehensive literature review should be conducted, and additional organisms may be included in the test set to strengthen the validity and broader applicability of the model.
3. The manuscript does not seem to incorporate state-of-the-art machine learning techniques, such as BERT or other large language models (LLMs), which have demonstrated significant success in tasks involving text analysis, including the analysis of amino acid sequences. I encourage the authors to explore and discuss the potential of these advanced models in their approach.
4. The manuscript lacks a thorough discussion of the existing models in the literature that serve the same purpose as TPGPred. The authors are encouraged to provide a comprehensive review of similar models in the introduction. Additionally, a direct comparison between TPGPred and currently available models using the same test set is essential for a fair evaluation and to demonstrate the viability and potential advantages and limitations of TPGPred.
5. The introduction lacks a clear explanation of the motivation behind developing a model like TPGPred. The authors are encouraged to provide a more comprehensive and detailed rationale, explaining the specific gaps in current research or practical applications that TPGPred aims to address.
6. Several mentions of supplementary material in the manuscript contain errors, such as 'Error! Reference source not found.' The authors should carefully review and correct these issues to ensure that all links to supplementary content are accurate and properly formatted.
Author Response
We are grateful to the editor and the reviewers for their efforts to handle and review our paper. All the comments have been addressed and any changes to the manuscript are highlighted in yellow for clarity.
Response to Reviewer 1 Comments
Comments and Suggestions for Authors
Comments 1. The test set appears to be disproportionately small compared to the training data, which may limit the robustness of the external validation. I recommend allocating at least 10% of the dataset for testing to ensure a more reliable and fair evaluation of the model's performance. Additionally, the test set should be stratified to ensure that the proportions of positive and negative class observations are consistent with those in the training data, which will help maintain representativeness and improve the validity of the results.
Response 1: Thank you for your suggestion. The test set size has been adjusted to 10%, with the positive and negative sample ratios in the test set proportionally balanced to match those in the training set. Additionally, all baseline models have been re-optimized using grid search to identify the optimal parameters for each model. Subsequently, all content, figures, and tables in the manuscript that involve model results have been updated accordingly. To better compare model performance, we included the results of three additional machine learning models, which are presented in Table S1 in the appendix. As their performance was suboptimal, they were not discussed in the main text.
Comments 2. The authors have only considered two organisms for building their models, which raises questions about the generalizability of the results. I suggest the authors provide a more thorough justification for this choice in the manuscript. Additionally, a comprehensive literature review should be conducted, and additional organisms may be included in the test set to strengthen the validity and broader applicability of the model.
Response 2: Thank you for your suggestion. We have revised the introduction, incorporating a comprehensive review of previous research and categorizing these studies for more detailed analysis. To better evaluate the adaptability of our model, we adjusted the external test samples and tested the model using both thermophilic and non-thermophilic proteins (Table 3). This approach allows for a more thorough assessment of the model's performance across different protein types.
To provide better clarity and explanation, we have added the following content to the introduction::
Numerous machine learning-based methods have been proposed and developed for the prediction and analysis of thermophilic proteins. Wang et al. [1, 2] utilized amino acid composition and g-gap dipeptides as protein feature encodings, employing a support vector machine (SVM) as the classifier. Feng et al. [3] analyzed the physicochemical properties of amino acids using an SVM classifier, focusing on a dataset with 500 positive and 500 negative samples for the classification of thermophilic proteins. Meng et al. [14] employed seven types of protein features as input and used SVM as the classifier. Tang et al. [4] used the frequency of amino acids and dipeptides as features and applied SVM for classification analysis on 1700 positive and negative samples. Guo et al. [5] analyzed ten protein encoding features using an SVM-based model, evaluating their effectiveness in classification. Charoenkwan et al. [6, 7] used 12 types of protein descriptors as input features and applied machine learning algorithms for feature analysis and classification. Zhao et al. [8] employed six types of protein encoding descriptors as input, further processing them with a convolutional neural network (CNN) and bidirectional long short-term memory network (BiLSTM), and used a multi-layer perceptron (MLP) for classification. Ahmed et al. [9] utilized seven protein descriptors as features and employed MLP as the classifier for protein classification, relying solely on protein descriptors without integrating non-linear sequence feature engineering methods. The aforementioned approaches primarily focus on the use of protein descriptors as the main feature set and analyze them using various machine learning methods. Pei et al. [10] directly used protein sequences as input, applying the Transformer-Embedding method within BERT for sequence feature processing, which is a pure sequence-based feature approach similar to word2vec and doc2vec. This model exclusively relies on a single sequence feature engineering method without performing a comparative analysis of different feature engineering strategies. Li et al. [11] used machine learning models based on protein sequences to study the potential maximum tolerance temperature of thermophilic proteins, achieving an R² of 0.75. Ahmed et al. [12] performed statistical analysis from a structural biology perspective, examining secondary structure, hydrogen bonds, salt bridges, DHA (donor-hydrogen-acceptor) angles, and bond lengths. These studies primarily focus on the structural and applied aspects of thermophilic protein analysis.
Overall, current methodologies predominantly focus on single categories of features, such as protein encoding or sequence-based characteristics. In the context of machine learning, features define the receptive field and scope through which the model perceives its target, serving as its primary source of information. Since machine learning methods are highly sensitive to the types of features employed, different feature engineering strategies often align with distinct, optimal machine learning algorithms. Rich and effective features are crucial for enhancing the performance of machine learning models. This study not only considers various feature engineering methods based on protein sequences but also emphasizes model interpretability by incorporating protein descriptor-based feature engineering techniques. The main contributions of this study to existing methodologies can be summarized as follows: (i) This study extensively employs protein encoding features, introducing a total of 14 types of protein descriptors, which enhances the interpretability of the model's results. (ii) The study incorporates a significant number of sequence-based features, using 11 different sequence feature engineering methods for comparison and analysis, which further improves model performance. (iii) In terms of model selection, seven machine learning models are analyzed, with four models discussed in detail in the main text, and the results of the other three models provided in the supplementary materials (Table_S 1). (iv) To further investigate the factors affecting model performance, this study explores four types of feature processing methods, three cross-validation techniques, and four feature selection methods, conducting comparative analyses to assess their impact on model efficacy. Through comprehensive comparative analyses, this study achieves further improvements in model performance.
Comments 3. The manuscript does not seem to incorporate state-of-the-art machine learning techniques, such as BERT or other large language models (LLMs), which have demonstrated significant success in tasks involving text analysis, including the analysis of amino acid sequences. I encourage the authors to explore and discuss the potential of these advanced models in their approach.
Response 3: Thank you for your suggestion. The Transformer-Embedding within BERT [10] can serve as a sequence feature engineering method for characterizing protein sequences. For comparative analysis, we have integrated this method into our study, with results for each model using this feature engineering approach labeled as "ModelName_BERT" in Table 2 and Table S1. These results can be directly compared with those from other feature engineering methods.
Regarding large language models (LLMs), we first attempted to test several existing LLMs by inputting corresponding protein sequences to assess their ability to recognize the given sequences. However, none of the tests yielded directly usable results (see Appendix 1).
Furthermore, most current LLMs are general-purpose models that excel in general domains. Many of these models have parameter counts exceeding 100 billion. Fine-tuning such large models for thermophilic protein classification would require a substantial amount of data, likely far beyond the scope of the dataset constructed for this study. Your suggestion is highly valuable, and we will further explore and analyze this approach in our future research.
Comments 4. The manuscript lacks a thorough discussion of the existing models in the literature that serve the same purpose as TPGPred. The authors are encouraged to provide a comprehensive review of similar models in the introduction. Additionally, a direct comparison between TPGPred and currently available models using the same test set is essential for a fair evaluation and to demonstrate the viability and potential advantages and limitations of TPGPred.
Response 4: Thank you for your suggestion. To better compare our work with existing studies, we have cited thirteen new relevant references in the introduction and conducted a categorized analysis. Our analysis of previous research reveals that most studies focus on a specific category, such as sequence-based string feature engineering methods or protein encoding descriptor-based feature engineering methods. Additionally, the scope and variety of analyses within each category are often limited. In this study, we employed eleven sequence-based string feature engineering methods, fourteen protein encoding descriptor-based feature engineering methods, seven machine learning models, four preprocessing algorithms, four feature selection methods, and three cross-validation methods. In total, forty-three different algorithms or models were involved, allowing for a comprehensive algorithmic analysis of thermophilic protein classification. Furthermore, we have supplemented and refined the external data test samples in Table 3, including new tests on non-thermophilic organisms, which further validate the effectiveness of our model.
To provide better clarity and explanation, we have added the following content to the introduction::
Numerous machine learning-based methods have been proposed and developed for the prediction and analysis of thermophilic proteins. Wang et al. [1, 2] utilized amino acid composition and g-gap dipeptides as protein feature encodings, employing a support vector machine (SVM) as the classifier. Feng et al. [3] analyzed the physicochemical properties of amino acids using an SVM classifier, focusing on a dataset with 500 positive and 500 negative samples for the classification of thermophilic proteins. Meng et al. [14] employed seven types of protein features as input and used SVM as the classifier. Tang et al. [4] used the frequency of amino acids and dipeptides as features and applied SVM for classification analysis on 1700 positive and negative samples. Guo et al. [5] analyzed ten protein encoding features using an SVM-based model, evaluating their effectiveness in classification. Charoenkwan et al. [6, 7] used 12 types of protein descriptors as input features and applied machine learning algorithms for feature analysis and classification. Zhao et al. [8] employed six types of protein encoding descriptors as input, further processing them with a convolutional neural network (CNN) and bidirectional long short-term memory network (BiLSTM), and used a multi-layer perceptron (MLP) for classification. Ahmed et al. [9] utilized seven protein descriptors as features and employed MLP as the classifier for protein classification, relying solely on protein descriptors without integrating non-linear sequence feature engineering methods. The aforementioned approaches primarily focus on the use of protein descriptors as the main feature set and analyze them using various machine learning methods. Pei et al. [10] directly used protein sequences as input, applying the Transformer-Embedding method within BERT for sequence feature processing, which is a pure sequence-based feature approach similar to word2vec and doc2vec. This model exclusively relies on a single sequence feature engineering method without performing a comparative analysis of different feature engineering strategies. Li et al. [11] used machine learning models based on protein sequences to study the potential maximum tolerance temperature of thermophilic proteins, achieving an R² of 0.75. Ahmed et al. [12] performed statistical analysis from a structural biology perspective, examining secondary structure, hydrogen bonds, salt bridges, DHA (donor-hydrogen-acceptor) angles, and bond lengths. These studies primarily focus on the structural and applied aspects of thermophilic protein analysis.
Overall, current methodologies predominantly focus on single categories of features, such as protein encoding or sequence-based characteristics. In the context of machine learning, features define the receptive field and scope through which the model perceives its target, serving as its primary source of information. Since machine learning methods are highly sensitive to the types of features employed, different feature engineering strategies often align with distinct, optimal machine learning algorithms. Rich and effective features are crucial for enhancing the performance of machine learning models. This study not only considers various feature engineering methods based on protein sequences but also emphasizes model interpretability by incorporating protein descriptor-based feature engineering techniques. The main contributions of this study to existing methodologies can be summarized as follows: (i) This study extensively employs protein encoding features, introducing a total of 14 types of protein descriptors, which enhances the interpretability of the model's results. (ii) The study incorporates a significant number of sequence-based features, using 11 different sequence feature engineering methods for comparison and analysis, which further improves model performance. (iii) In terms of model selection, seven machine learning models are analyzed, with four models discussed in detail in the main text, and the results of the other three models provided in the supplementary materials (Table_S 1). (iv) To further investigate the factors affecting model performance, this study explores four types of feature processing methods, three cross-validation techniques, and four feature selection methods, conducting comparative analyses to assess their impact on model efficacy. Through comprehensive comparative analyses, this study achieves further improvements in model performance.
Comments 5. The introduction lacks a clear explanation of the motivation behind developing a model like TPGPred. The authors are encouraged to provide a more comprehensive and detailed rationale, explaining the specific gaps in current research or practical applications that TPGPred aims to address.
Response 5: Thank you for your suggestion. In terms of methodology, our literature review reveals that existing research tends to focus on specific types of features without effectively integrating multiple feature categories. Moreover, the range of features explored within a single category is often limited, lacking a broader analysis and selection of features. In this study, we combine sequence-based string feature engineering methods with protein encoding descriptor-based feature engineering approaches. This integrated feature set is then used to optimize and test various machine learning models, aiming to achieve effective classification of thermophilic proteins.
The study of thermophilic proteins holds significant potential for reducing the energy consumption of cooling systems in large-scale fermenters. In industrial applications, transitioning from laboratory research to industrial-scale production presents challenges due to the substantial heat generated during microbial fermentation processes. To maintain optimal growth conditions for microorganisms, large cooling systems are required to dissipate the excess heat generated, ensuring that the fermenter operates within a suitable temperature range. Identifying thermophilic proteins and employing synthetic biology approaches to express these proteins in industrial microbial chassis can transform engineered strains into thermotolerant variants. Such modifications are advantageous for achieving cost-effective, large-scale fermentation of target products.
Therefore, the primary goal of this study is to identify thermophilic proteins and utilize them to modify chassis strains, thereby reducing the operating time and intensity of cooling systems, ultimately lowering energy consumption during large-scale fermentation processes. To elucidate this concept, we have expanded the introduction with a detailed literature review and analysis. Additionally, we have included the following statement: "In the transition from laboratory research to industrial-scale production, the heat generated during microbial fermentation necessitates large cooling systems to remove excess heat and maintain optimal growth conditions for microorganisms. Identifying thermophilic proteins and using synthetic biology techniques to express suitable thermophilic proteins in industrial microbial chassis can transform engineered strains into thermotolerant variants, facilitating cost-effective, large-scale production of target compounds.”
Comments 6. Several mentions of supplementary material in the manuscript contain errors, such as 'Error! Reference source not found.' The authors should carefully review and correct these issues to ensure that all links to supplementary content are accurate and properly formatted.
Response 6: Thank you for your suggestion. We have scrutinized and revised this section as suggested.
Appendix 1:LLMs test:
chatGPT4o:https://chatgpt.com/
prompt:
MKRWLLAVGVALGLGALAQTGKLEIFSWWAGDEGPALEALIKLYKQKYPGVEVINATVTGGAGGNAKAVLKTRMLGGNPPDTFQVHAGMELIGTWVVADRMEDLTPLFRQEGWLKAFPKGLLDLISYKGGIWSVPVNIHRSNVMWYMPAKLKEWGVTPPKTWAEFLATCQTLKKKGLEAPLALGENWTQQHLWESVALGILGPEGWNNLWNGKLKFTDPKAVAVWETFGKVLDCANKDAAGLSWQQAVDRVVQGQAAFNVMGDWAAGYMSTTLKLKPGTDFAWAPSPGTSGVFMMLSDSFGLPKGAKNRQNAINWLKLVGSKEGQDTFNPLKGSIAARLDSDPAKYNAYGQSAMKDWQSNRIVGSLVHGAVAPESFMSQFGTVMEIFLQSKNPQAAANAAQAIANQVGLGR,Is this a thermophilic protein?
Answer:
To determine whether this protein is thermophilic, we need to analyze its properties based on amino acid composition and certain structural features that are typically associated with thermophilic proteins.
Key Characteristics of Thermophilic Proteins:
Increased Arginine (R) and Tyrosine (Y) content: These amino acids help stabilize the protein through salt bridges and hydrogen bonding, contributing to higher thermal stability.
Decreased Glutamine (Q) and Asparagine (N) content: These amino acids tend to destabilize proteins at high temperatures due to their tendency to undergo deamination.
Additional features: Thermophilic proteins often have increased hydrophobic interactions, compact cores, and more salt bridges or hydrogen bonds.
deepseek:https://chat.deepseek.com/
prompt:
MKRWLLAVGVALGLGALAQTGKLEIFSWWAGDEGPALEALIKLYKQKYPGVEVINATVTGGAGGNAKAVLKTRMLGGNPPDTFQVHAGMELIGTWVVADRMEDLTPLFRQEGWLKAFPKGLLDLISYKGGIWSVPVNIHRSNVMWYMPAKLKEWGVTPPKTWAEFLATCQTLKKKGLEAPLALGENWTQQHLWESVALGILGPEGWNNLWNGKLKFTDPKAVAVWETFGKVLDCANKDAAGLSWQQAVDRVVQGQAAFNVMGDWAAGYMSTTLKLKPGTDFAWAPSPGTSGVFMMLSDSFGLPKGAKNRQNAINWLKLVGSKEGQDTFNPLKGSIAARLDSDPAKYNAYGQSAMKDWQSNRIVGSLVHGAVAPESFMSQFGTVMEIFLQSKNPQAAANAAQAIANQVGLGR,Is this a thermophilic protein? Please answer in English.
answer:
The provided protein sequence does not exhibit clear characteristics that are typically associated with thermophilic proteins. Thermophilic proteins are known for having specific features that enhance their stability at high temperatures, such as increased hydrophobic interactions, higher proline content, and specific amino acid substitutions that promote thermal stability.
The sequence provided does not show a significant number of these characteristic features. For example, it does not have an unusually high proline content or other specific amino acid substitutions that are commonly found in thermophilic proteins. Therefore, based on the sequence alone, it is unlikely to be a thermophilic protein.
References
- Wang, X.-F.; Lu, F.; Du, Z.-Y.; Li, Q.-M., Prediction model of thermophilic protein based on Stacking Method. Current Bioinformatics 2021, 16, (10), 1328-1340.
- Wang, X.-F.; Gao, P.; Liu, Y.-F.; Li, H.-F.; Lu, F., Predicting thermophilic proteins by machine learning. Current Bioinformatics 2020, 15, (5), 493-502.
- Feng, C.; Ma, Z.; Yang, D.; Li, X.; Zhang, J.; Li, Y., A method for prediction of thermophilic protein based on reduced amino acids and mixed features. Frontiers in bioengineering and biotechnology 2020, 8, 285.
- Tang, H.; Cao, R.-Z.; Wang, W.; Liu, T.-S.; Wang, L.-M.; He, C.-M., A two-step discriminated method to identify thermophilic proteins. International Journal of Biomathematics 2017, 10, (04), 1750050.
- Guo, Z.; Wang, P.; Liu, Z.; Zhao, Y., Discrimination of thermophilic proteins and non-thermophilic proteins using feature dimension reduction. Frontiers in Bioengineering and Biotechnology 2020, 8, 584807.
- Charoenkwan, P.; Schaduangrat, N.; Moni, M. A.; Manavalan, B.; Shoombuatong, W., SAPPHIRE: A stacking-based ensemble learning framework for accurate prediction of thermophilic proteins. Computers in Biology and Medicine 2022, 146, 105704.
- Charoenkwan, P.; Chotpatiwetchkul, W.; Lee, V. S.; Nantasenamat, C.; Shoombuatong, W., A novel sequence-based predictor for identifying and characterizing thermophilic proteins using estimated propensity scores of dipeptides. Scientific Reports 2021, 11, (1), 23782.
- Zhao, J.; Yan, W.; Yang, Y., DeepTP: a deep learning model for thermophilic protein prediction. International Journal of Molecular Sciences 2023, 24, (3), 2217.
- Ahmed, Z.; Zulfiqar, H.; Khan, A. A.; Gul, I.; Dao, F.-Y.; Zhang, Z.-Y.; Yu, X.-L.; Tang, L., iThermo: a sequence-based model for identifying thermophilic proteins using a multi-feature fusion strategy. Frontiers in Microbiology 2022, 13, 790063.
- Pei, H.; Li, J.; Ma, S.; Jiang, J.; Li, M.; Zou, Q.; Lv, Z., Identification of thermophilic proteins based on sequence-based bidirectional representations from transformer-embedding features. Applied Sciences 2023, 13, (5), 2858.
- Li, M.; Wang, H.; Yang, Z.; Zhang, L.; Zhu, Y., DeepTM: A deep learning algorithm for prediction of melting temperature of thermophilic proteins directly from sequences. Computational and Structural Biotechnology Journal 2023, 21, 5544-5560.
- Ahmed, Z.; Zulfiqar, H.; Tang, L.; Lin, H., A statistical analysis of the sequence and structure of thermophilic and non-thermophilic proteins. International Journal of Molecular Sciences 2022, 23, (17), 10116.

Reviewer 2 Report
Comments and Suggestions for Authors
Review of "TPGPred: A Mixed-Feature-Driven Approach for Identifying Thermophilic Proteins Based on Gradient Boosting"
The manuscript describes the development of a machine learning model, TPGPred, designed to predict thermophilic proteins using gradient boosting and mixed feature engineering. With its insights into the identification of thermophilic proteins—which have important biotechnological applications—the study makes a substantial addition to the field.
Strengths:
- The authors built a well-curated dataset for thermophilic and non-thermophilic proteins.
- The combination of string-based methods with 14 protein descriptors appears to enhance the model’s predictive performance.
- The use of multiple machine learning techniques and metrics like AUROC, AUPRC, and MCC provides a well-rounded evaluation.
Recommendations:
-How were data biases minimized in the selection of thermophilic vs. non-thermophilic proteins? Could more details be provided on the criteria used for selecting sequences?
-The manuscript discusses the use of SHAP values to assess feature importance. Could specific examples of features most relevant to thermophilic protein prediction be provided? How might these features relate to the known biology of thermophilic proteins?
- Why was Gradient Boosting chosen as the baseline model over other potential models, like deep neural networks or support vector machines? Could a brief justification or comparison of initial model trials be added?
- Could specific biological insights or hypotheses about the structural characteristics of thermophilic proteins derived from feature analysis be added?
- SMOTE was used to address class imbalance, were alternative approaches like cost-sensitive learning considered? How did different resampling techniques compare in terms of impact on performance metrics?
- The authors tested TPGPred on an independent test set. Were additional datasets from diverse species or environmental conditions used to validate the model's generalizability?
Conclusion:
The manuscript presents a solid foundation for thermophilic protein prediction using machine learning. Addressing these issues mentioned above could improve the paper's impact and clarity.
Author Response
We are grateful to the editor and the reviewers for their efforts to handle and review our paper. All the comments have been addressed and any changes to the manuscript are highlighted in yellow for clarity.
Response to Reviewer 2 Comments
Comments1. How were data biases minimized in the selection of thermophilic vs. non-thermophilic proteins? Could more details be provided on the criteria used for selecting sequences?
Response 1: Thank you for your suggestion. First, prior research is necessary to confirm whether the strain is an extreme thermophile or a true non-thermophilic bacterium. In this study, the sequence of the non-thermophilic strain was obtained through two rounds of sequencing, and the resulting data have been made publicly available. Additionally, we have conducted extensive studies and modifications on this strain, leading to a high degree of confidence in its classification as non-thermophilic.
Second, the protein sequences were subjected to rigorous analysis. Sequences containing characters outside of those specified in the [ARNDCQEGHILKMFPSTWYV] list were removed. Furthermore, sequences containing non-alphabetic symbols, such as [-!@#ï¿¥%……&*-], empty sequences, and those of insufficient length (e.g., shorter than 20) were excluded from the analysis.
To explain this, we have modified Section 3.1 by adjusting and adding the sentences “During the dataset construction process, we excluded invalid amino acid sequences and conducted a thorough analysis of the selected sequences based on their composition, length, and symbols. Specifically, we filtered out sequences containing non-alphabetic characters, sequences with abnormal markers, invalid sequences, sequences shorter than 20 residues, and empty sequences.”
Comments 2. The manuscript discusses the use of SHAP values to assess feature importance. Could specific examples of features most relevant to thermophilic protein prediction be provided? How might these features relate to the known biology of thermophilic proteins?
Response 2: Thank you for your suggestion. The protein descriptors designed in this study possess biological significance, with each descriptor comprising multiple sub-features. The specific number of sub-features corresponding to each descriptor is as follows:
["AAC": 20, "CKSAAP": 2400, "CTDC": 39, "CTDT": 39, "CTDD": 195, "CTriad": 343, "DPC": 400, "GAAC": 5, "GDPC": 25, "CKSAAGP": 150, "DDE": 400, "GTPC": 125, "KSCTriad": 343, "TPC": 8000].
As shown in Figure 11, the most relevant protein descriptor identified is GAAC.
Example output for GAAC and its sub-features:
feature name |
Sub-feature name |
importance |
GAAC |
uncharge |
0.23415 |
GAAC |
alphatic |
0 |
GAAC |
aromatic |
0 |
GAAC |
postivecharge |
0 |
GAAC |
negativecharge |
0 |
GAAC (Global Amino Acid Composition) is a feature-based method that describes the proportion of each amino acid within a sequence by calculating the frequency of individual amino acids. This approach provides insights into the overall amino acid composition of a protein. Thermophilic proteins often contain a higher proportion of stabilizing amino acids, such as arginine (Arg) and glutamate (Glu), which are capable of forming strong hydrogen bonds and ionic interactions, thereby enhancing protein stability under high-temperature conditions.
The uncharged amino acids, refer to those amino acids with side chains that do not carry a charge, such as glycine (Gly), alanine (Ala), and serine (Ser). In thermophilic proteins, the proportion of uncharged amino acids may be linked to protein stability. For instance, nonpolar amino acids contribute to the formation of hydrophobic cores, which are critical for maintaining protein stability in high-temperature environments. Hydrophobic interactions play a vital role in preserving protein structure at elevated temperatures. Thus, thermophilic proteins require not only adequate hydrophobic interactions to prevent thermal denaturation but also rely on ionic bonds and hydrogen bonds. As a result, the balance between charged and uncharged amino acids directly influences the stability of proteins under thermal stress.
To explain this, we have modified Section 3.10 by adjusting and adding the sentences “The uncharge sub-feature from GAAC (Global Amino Acid Composition) had a weight of 0.23415, ranking second among the features, indicating that the global composition of uncharged amino acids significantly influences the model's predictions. From a biological and structural perspective, uncharged amino acids play a critical role in forming the hydrophobic core of proteins. The proportion of uncharged residues is likely associated with protein stability."
Comments 3. Why was Gradient Boosting chosen as the baseline model over other potential models, like deep neural networks or support vector machines? Could a brief justification or comparison of initial model trials be added?
Response 3: Thank you for your suggestion. In this study, no baseline model was initially established in Section 3.2 and Table 2. Instead, we evaluated various sequence-based string feature engineering methods across four different models, aiming to identify the most suitable method for each machine learning model. Subsequently, in Section 3.4 and Figures 2 to 7, we sought to determine the optimal set of protein descriptors for each model among 14 types of descriptors. Following this, we assessed the overall performance of each model combined with the best-matching string feature engineering method and protein descriptor set. The model-feature combination with the highest overall performance was then selected as the foundational model for further comparisons. Using this foundational model, we conducted an in-depth analysis of how different feature processing methods, feature selection techniques, and cross-validation strategies impacted the final results.
To explain this, we have modified Section 3.3 and 3.5 by adjusting and adding the sentences “In the initial analysis phase, no specific model was designated as the baseline model for comparison. Instead, the focus was on identifying the most suitable feature engineering methods for each machine learning model. The aim was to explore the optimal combinations of feature engineering techniques and models without any preconceived preferences.” “Combining each machine learning model with its most compatible string feature engineering method and protein descriptor set enables a more comprehensive performance evaluation. This approach allows for a deeper analysis of the overall advantages of different model-feature combinations, facilitating a more nuanced comparison.”
Comments 4. Could specific biological insights or hypotheses about the structural characteristics of thermophilic proteins derived from feature analysis be added?
Response 4: Thank you for your suggestion. From a biological perspective, the hydrophobic structure of proteins plays a critical role in the recognition of thermophilic proteins. For example, the high-weight uncharged amino acids in the GAAC descriptor (e.g., glycine and alanine) contribute to the formation of the hydrophobic core within proteins. In the AAC descriptor, isoleucine (I) ranks highly, with a weight of 0.025993. As a hydrophobic amino acid, isoleucine helps to strengthen internal hydrophobic interactions within proteins, thus maintaining their folded structure under high-temperature conditions. The dipeptide patterns QL (glutamine-leucine) and EQ (glutamic acid-glutamine) from the DDE (Dipeptide Deviation from Expected mean) descriptor may be involved in local hydrogen bonding and polar interactions, which can help stabilize protein structures locally at elevated temperatures. Additionally, the “hydrophobicity_CASG920101.G3” feature from the CTDC descriptor, with a weight of 0.017529, indicates that physicochemical properties related to hydrophobicity play a role in the model's predictive capabilities.
To explain this, we have modified Section 3.10 by adding the sentences “From a biological perspective, the hydrophobic structure of proteins plays a important role in the identification of thermophilic proteins. For example, uncharged amino acids (e.g., glycine and alanine) in the GAAC descriptor, isoleucine (I) in the AAC descriptor, dipeptide patterns such as QL (glutamine-leucine) and EQ (glutamic acid-glutamine) in the DDE (Dipeptide Deviation from Expected mean) descriptor, and the “hydrophobicity_CASG920101.G3” feature in the CTDC descriptor, these features, all related to hydrophobicity, contribute significantly to the model’s predictive performance.”
Comments 5. SMOTE was used to address class imbalance, were alternative approaches like cost-sensitive learning considered? How did different resampling techniques compare in terms of impact on performance metrics?
Response 5: Thank you for your suggestion. In Figure 8, we have added the results of the cost-sensitive learning approach. Through comparison, we found that the cost-sensitive learning algorithm may be more suitable for scenarios with a substantial imbalance between positive and negative samples.
To explain this, we have modified Section 3.6 by adding the sentences “In contrast, the cost-sensitive algorithm performs noticeably worse than other algorithms across most metrics. This approach may be more suitable for scenarios where there is a significant imbalance between positive and negative samples.”
Comments 6. The authors tested TPGPred on an independent test set. Were additional datasets from diverse species or environmental conditions used to validate the model's generalizability?
Response 6: Thank you for your suggestion. The strains in the independent test set presented in Table 3 differ from those used in the training set, each possessing distinct functions and applications. However, they may exhibit similar thermophilic and non-thermophilic characteristics. To demonstrate the model's ability to discriminate between positive and negative samples, we adjusted the composition of the independent test set in Table 3 by adding more non-thermophilic strains. Additionally, we provided detailed information including strain names, accession numbers, and references.
To explain this, we have modified Section 3.9 by adding the sentences “Table 3 presents the results of testing thermophilic proteins referenced in the literature. These strains are distinct from those used in the training and test sets, each possessing specific functions and applications. However, they share similar thermophilic and non-thermophilic characteristics with the strains used in the training and test sets.”

Round 2
Reviewer 1 Report
Comments and Suggestions for Authors
The recent revisions to the manuscript have greatly improved its clarity and depth. Nonetheless, I recommend one final adjustment, which is detailed below.
Question 1.
When answering Question 3, the authors seem to have missed the point regarding the use of large language models (LLMs) in their study. My original suggestion was aimed at encouraging the application of advanced models like BERT for the analysis of amino acid sequences, not simply testing GPT-like models.While I appreciate the mention of the Transformer-Embedding within BERT and its integration into the study, the experiments described in Appendix 1 do not fulfill the original intent of the comment. The results from testing existing LLMs with protein sequences, which ultimately did not yield usable outcomes, do not align with the recommendation to utilize models specifically designed for text analysis tasks, such as BERT. Consequently, the authors are requested to remove Appendix 1 or any mention of it from the manuscript, as it does not contribute to the discussion on the application of state-of-the-art machine learning techniques in the context of this study. Since the authors utilized a BERT-based model, my original request has been satisfactorily addressed.
Author Response
We are grateful to the editor and the reviewers for their efforts to handle and review our paper. All the comments have been addressed and any changes to the manuscript are highlighted in yellow for clarity.
Comments 1: When answering Question 3, the authors seem to have missed the point regarding the use of large language models (LLMs) in their study. My original suggestion was aimed at encouraging the application of advanced models like BERT for the analysis of amino acid sequences, not simply testing GPT-like models.While I appreciate the mention of the Transformer-Embedding within BERT and its integration into the study, the experiments described in Appendix 1 do not fulfill the original intent of the comment. The results from testing existing LLMs with protein sequences, which ultimately did not yield usable outcomes, do not align with the recommendation to utilize models specifically designed for text analysis tasks, such as BERT. Consequently, the authors are requested to remove Appendix 1 or any mention of it from the manuscript, as it does not contribute to the discussion on the application of state-of-the-art machine learning techniques in the context of this study. Since the authors utilized a BERT-based model, my original request has been satisfactorily addressed.
Response 1: We sincerely appreciate the reviewers' insightful suggestions. In response, we have thoroughly reviewed the manuscript and have removed all references to LLMs. Additionally, we have refrained from incorporating any LLM-related content in Appendix 1.